# Don't Run with Scissors ✂: Pruning Breaks VLA Models but They Can Be Recovered

## Abstract

Vision–Language–Action (VLA) models have advanced robotic capabilities but remain challenging to deploy on resource-limited hardware. Pruning has enabled efficient compression of large language models (LLMs), yet it is largely understudied in robotics. Surprisingly, we observe that pruning VLA models leads to drastic degradation and increased safety violations. We introduce **GLUESTICK**, a post-pruning recovery method that restores much of the original model's functionality while retaining sparsity benefits. Our method performs a one-time interpolation between the dense and pruned models in weight-space to compute a corrective term. This correction is used during inference by each pruned layer to recover lost capabilities with minimal overhead. GLUESTICK requires no additional training, is agnostic to the pruning algorithm, and introduces a single hyperparameter that controls the tradeoff between efficiency and accuracy. Across diverse VLA architectures and tasks in manipulation and navigation, GLUESTICK achieves competitive memory efficiency while substantially recovering success rates and reducing safety violations. Videos, code, and additional materials are in: **https://gluestick-vla.github.io/**.

**Task:** Enter the dining room and walk to the other end.     **Task:** Put the bowl on top of the cabinet.

⬅ **Pruned VLA** ✂     ⬅ **GLUESTICK** 🖍

Figure 1: **VLAs break under pruning, and GLUESTICK fixes them.** Pruning methods unexpectedly cause task and safety failures in VLAs: colliding with an object in a navigation task **(left)**, or dropping a bowl in a manipulation task **(right)**. Our post-pruning method, GLUESTICK, restores the lost functionality of the original model.

## 1 Introduction

Vision-Language-Action (VLA) models mark a new era in robotics. Earlier approaches to robot control used pipelines that separated perception, planning, and control into distinct subsystems. VLAs instead integrate these components into a single end-to-end framework, leveraging large language models (LLMs) to connect perception and natural language instructions directly to action (Kim et al., 2024; Li et al., 2024; Lee et al., 2025; Black et al., 2024; Bjorck et al., 2025; Brohan et al., 2022; Zitkovich et al., 2023). VLAs learn generalized action policies from internet-scale robotics data, enabling them to transfer across diverse

tasks and environments (O'Neill et al., 2024). VLAs can also take advantage of pretrained vision and language models, giving them rich semantic knowledge while grounding behavior in real-world observations (Achiam et al., 2023; Team et al., 2023; Betker et al., 2023).

The growing capabilities of VLAs come at a cost. As in LLMs, VLAs follow a scaling trend wherein their capabilities grow as the size of the model grows larger (Kaplan et al., 2020). In robotics, this scaling is especially consequential because deployment typically occurs on hardware with strict limits on memory, power, and throughput. For example, an industry-standard Jetson Orin NX provides only 8–16GB of shared CPU-GPU memory (Liu et al., 2024b; Rey et al., 2025), far below server-grade GPUs used for foundation models, such as the NVIDIA HGX B200 with 180GB of memory (NVIDIA, 2025). Such limitations make *compression* necessary to fit models on resource-constrained hardware. Yet a clear gap remains in understanding how efficiency gains from compression intersect with VLA model success and safety—*a gap this work directly seeks to answer, specifically about pruning.*

*Pruning* is a key compression technique for large language models (Frantar & Alistarh, 2023; Sun et al., 2023a;b). It produces smaller models and more efficient GPU execution with optimized sparse kernels by removing unnecessary weights and enforcing structured sparsity. However, we surprisingly observe that pruning introduces unique challenges for VLAs. Whereas pruning techniques are effective for LLMs, applying the same methods to VLAs leads to catastrophic degradation. In our experiments, recent pruning algorithms reduced task success rate on manipulation tasks from 85.2% to 0.0% and on a navigation task from 43.0% to 0.0%, while also increasing the frequency of safety violations.

We recover success rates and reduce safety violations by introducing **GLUESTICK**, a new post-pruning recovery method that recovers signal lost during pruning while preserving the efficiency benefits of sparsity. GLUESTICK operates entirely in weight space, using the information discarded by pruning to nudge the model back toward more performant regions without any retraining. This is achieved by adding a lightweight correction term, computed from singular values in the gap between the dense and pruned weights.

Our approach is pruning-agnostic and can be applied on top of any existing pruning algorithm. In doing so, GLUESTICK restores up to 100% of performance in navigation tasks and as much as 60% in dexterous manipulation domains, while maintaining the efficiency gains of structured sparsity (Figure 1). Finally, GLUESTICK introduces a single interpretable hyperparameter that allows practitioners to directly control the trade-off between accuracy and efficiency, making it adaptable to diverse application requirements. We demonstrate that our method consistently recovers performance and improves safety across three VLA architectures, two widely used robotics benchmarks, and multiple robot embodiments.

**Our contributions.**

- **Empirical evidence of pruning collapse in VLAs.** We present the first systematic study showing that pruning, which is effective for LLMs, causes near-complete collapse of the success rate in embodied VLA models, and an increase in safety violations.

- **Study of why VLAs differ from LLMs under pruning.** Through spectral analysis, we identify structural properties of VLA architectures that could make them more fragile to pruning than language-only models.

- **An effective, training-free recovery method.** We propose GLUESTICK, a post-pruning recovery algorithm that restores lost signal after pruning using a low-rank, lightweight correction in weight space. GLUESTICK is pruning-algorithm-agnostic, requires no retraining, and introduces only a single interpretable hyperparameter.

## 2 RELATED WORK & MOTIVATION

**VLA Models.** Recent work has focused on developing VLA models that unify perception, language understanding, and decision-making into a single policy mapping multimodal inputs directly to robot actions. These models typically consist of three components: a vision backbone, a multimodal projector, and a language backbone. For instance, given image observations and natural language instructions, OpenVLA (Kim et al., 2024) outputs end-

| Model | Method | Succ. | Unsafe |
|-------|--------|-------|--------|
| OpenVLA | Dense | 85.2% | 33.4% |
| | Magnitude | 0.0% | 46.4% |
| | Wanda | 0.0% | 51.6% |
| NaVILA | Dense | 43.0% | 23.0% |
| | Magnitude | 0.0% | 100.0% |
| | Wanda | 0.0% | 46.0% |

Table 1: **Success and unsafe-episode rate across pruning strategies.** Succ.=% successful episodes; Unsafe=% with a safety violation.

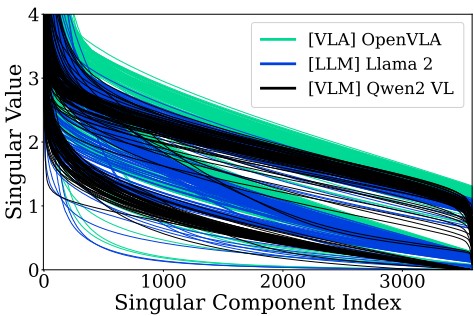

Figure 2: **Singular value spectra of weights**. VLA vs. LLM/VLM showing that VLA spectra are flatter and signal is more dispersed across the weight space.

effector poses and gripper commands for manipulation, while NaVILA (Cheng et al., 2025a) generates velocity commands for navigation. Other prominent systems in this space include RT (Brohan et al., 2022; Zitkovich et al., 2023), the $\pi$ series (Black et al., 2024; Intelligence et al., 2025), PaLM-E (Driess et al., 2023), Gr00t N1 (Bjorck et al., 2025), and CogACT (Li et al., 2024), all of which share the commonality of being large end-to-end transformer-based policies with billions of parameters. Their size poses a particular challenge for robotics, which have tight resource contraints (Jabbour & Janapa Reddi, 2024), making compression techniques such as pruning especially important for deployment. Exacerbating this challenge, there is a clear trend toward richer inputs and outputs: for example, SpatialVLA (Qu et al., 2025) incorporates not only image token inputs but also 3D scene information, while MolmoAct (Lee et al., 2025) and WorldVLA (Cen et al., 2025) extend outputs beyond action vectors to include depth predictions or full world models. These expansions further grow model size and demand, underscoring the importance of studying efficiency–success trade-offs in compressed VLA models for practical robotic deployment.

**Pruning Benefits for Robotics.** Pruning is a common technique for compressing LLMs, where a fraction of weights are set to zero (Zhu et al., 2024). Magnitude (Han et al., 2015) and Wanda (Sun et al., 2023b) are widely used pruning methods, valued for being training-free and computationally efficient. Magnitude removes small-magnitude weights, while Wanda scores connections by activation statistics on calibration inputs and prunes those deemed less important. Pruning reduces both parameter count and FLOPs while often preserving accuracy, and is especially effective when applied in hardware-friendly patterns such as structured "N:M" sparsity (e.g., 2:4). Modern GPUs exploit these patterns with specialized kernels that reduce memory traffic and multiply-accumulate operations (MACs) (Cheng et al., 2025b); for example, NVIDIA's Sparse Tensor Cores and cuSPARSELt accelerate 2:4 sparse general matrix multiplications (Mishra et al., 2021). The reduced computation due to pruning not only enables acceleration and memory savings but also significantly cuts power consumption (Han et al., 2016). These benefits are especially attractive for robotics, where devices operate under strict constraints on compute, memory, and energy, making it critical to also understand pruning's broader impact on VLA model success and safety.

**Pruning Impacts.** When pruning methods are applied to LLMs they achieve strong accuracy retention. On LLaMA-2-70B (Touvron et al., 2023), mean accuracy on the EleutherAI LM Harness (Gao et al., 2021) decreases by only 7.13% with Magnitude pruning and 2.94% with Wanda (Sun et al., 2023b), despite imposing a strict 2:4 structured sparsity on the original model and removing 50% of weights. More recent work has investigated the impact of pruning on Vision-Language Models (VLMs) and has shown a larger accuracy decrease compared to LLMs. For example, Koike-Akino et al. (2025) reported that on the ScienceQA dataset, LLaVA-7B experienced a 12.5% accuracy drop after being pruned with Wanda at 50% sparsity on image-based tasks. Similarly, Liang et al. (2025) observed a 9%–30% accuracy drop in LLaVA-SQA-7B and LLaVA-v1.5-7B. These foundational pieces of work illustrate that pruning has a measurable impact on VLMs, and our work builds on this observation to explore whether this trend continues or changes for VLA models. Taken

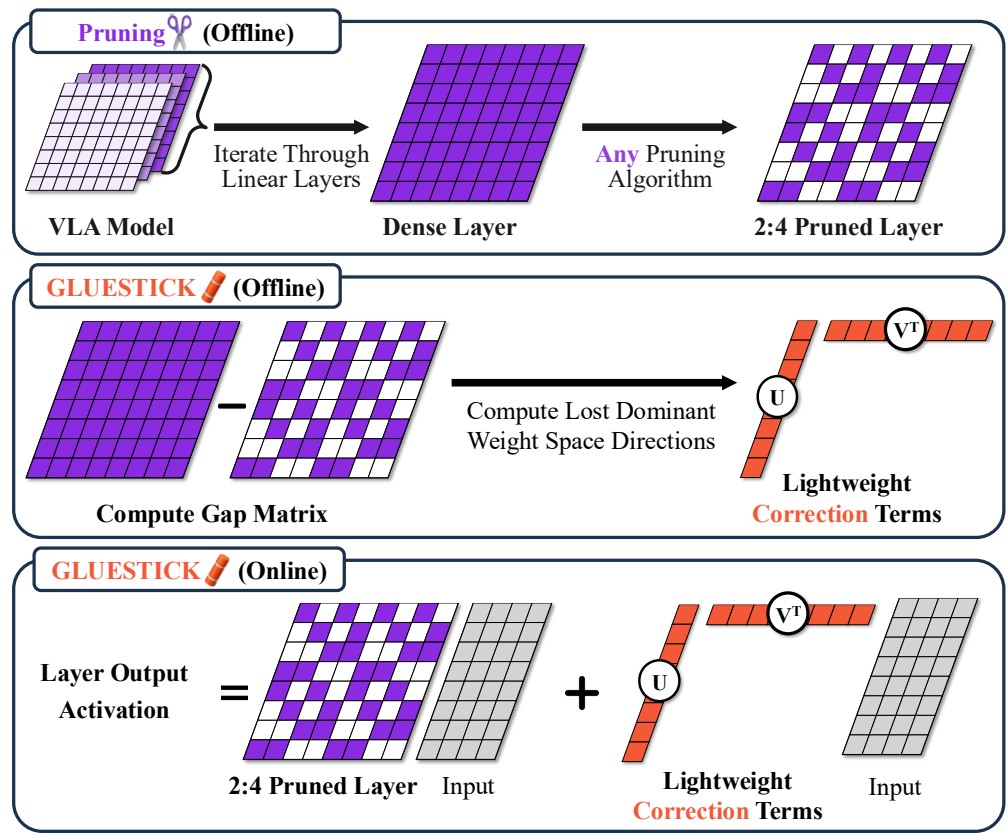

Figure 3: **Overview of GLUESTICK. (Top)** A VLA model is pruned with a standard algorithm (e.g., Wanda) to enforce 2:4 sparsity in linear layers. **(Middle)** Offline, we compute the gap between the dense and pruned weights and extract dominant lost directions via SVD, yielding lightweight corrections. **(Bottom)** At inference time, these correction terms are applied alongside the pruned weights, effectively adding back lost signal.

together, this line of work suggests that while pruning has been validated on LLMs, it can lead to greater degradation in VLM performance. Therefore, weight adjustments cannot be assumed to be benign in all contexts. In robotics, where VLAs must balance efficiency with both task success and safety, the implications of pruning remain unexplored. No prior work has examined how pruning affects VLA success or safety and how such impacts might be recovered. Direct weight-space interventions for post-pruned VLAs are similarly underexplored. In this paper, we address this gap by analyzing pruning's effect on VLA models and introducing a training-free recovery method that restores both success and safety.

## 3    GLUESTICK

In this section, we first present our surprising finding that pruning can cause catastrophic degradation in VLA success and safety. We then introduce our new method, **GLUESTICK** (sin**G**ular va**LUE STIC**hing), which glues pruned VLAs back to high task success rates and safe behaviors as in the original dense models. Additional details and pseudocode are available in Appendix A.

### 3.1    Impact of Pruning on VLA Models

Pruning has substantially reduced memory usage in language models, with minimal loss in accuracy (see Section 2). At the same sparsity levels commonly used in LLMs, we observe surprisingly different outcomes on popular VLAs. Representative VLAs such as Open-

VLA (Kim et al., 2024) and NaVILA (Cheng et al., 2025a), when pruned with Magnitude or Wanda, drop in success rate to 0% (from 85.2% and 43.0%, respectively). These come alongside a rise in their unsafe-episode rates; in the worst case OpenVLA increases from 33.4% to 51.6%, and for NaVILA from 23.0% to 100.0% (see Table 1). These findings show that text-validated pruning does not directly transfer to embodied control.

To explain why pruning degrades VLAs far more than LLMs or VLMs, we ask whether their weight-space properties differ. Thus, we examine the singular value spectra of equal-sized layers within the language backbones of OpenVLA (VLA), LLaMA-2-7B (LLM) (Touvron et al., 2023), and Qwen2-VL-Instruct-7B (VLM) (Bai et al., 2025). We display these spectra in Figure 2, where each line corresponds to the singular values of a single layer, obtained by computing the SVD of that layer's weight matrix and plotting its full set of singular values. In LLM and VLM models, the spectra are more anisotropic, evidenced by a steep initial drop followed by a long tail, which concentrates energy in a few dominant directions. This profile helps pruning, since removing small coefficients mainly trims low-energy directions while leaving the principal subspaces intact. In contrast, VLA layers show a noticeably flatter decay, indicating energy spread across many directions. In this regime, even small-magnitude coefficients contribute to important subspaces, so pruning discards useful signal distributed throughout the matrix. Based on this insight, our method in the following section explores the recovery of this lost information within the weight space.

### 3.2 GLUESTICKING PRUNED MODELS

The space of pruning configurations is combinatorial, making optimal selection of weights to remove intractable. Heuristic methods such as Magnitude and Wanda sidestep the global optimization by scoring individual weights and pruning by score. While simple and efficient, these heuristics discard correlated weights under grouped sparsity constraints (e.g., 50% sparsity with 2:4 or 4:8 groups), which could be especially harmful for VLA models.

We propose **GLUESTICK**, a post-hoc, training-free recovery method that operates entirely in weight space and is agnostic to the pruning algorithm (see Figure 3). GLUESTICK requires only the original dense model and its pruned counterpart, and incurs a one-time offline cost; no additional training is required.

Specifically, for each linear layer with dense weight matrix $W_{\text{dense}} \in \mathbb{R}^{d_{\text{out}} \times d_{\text{in}}}$ and its pruned version $W_{\text{pruned}}$ (fixed, preserving the original 2:4/4:8 pattern), we define the *gap matrix*:

$$W_{\text{gap}} = W_{\text{dense}} - W_{\text{pruned}}, \tag{1}$$

which captures lost information due to pruning. We then compute a truncated singular value decomposition (SVD) of the gap matrix:

$$W_{\text{gap}} = U\Sigma V^\top \approx U_r \Sigma_r V_r^\top, \tag{2}$$

keeping the top $r$ singular components. By Eckart & Young (1936), this is the *best rank-$r$* approximation to $W_{\text{gap}}$ in Frobenius norm. For memory and speed, we fold the singular values into one term so that only two compact matrices need to be stored:

$$A = U_r \Sigma_r \in \mathbb{R}^{d_{\text{out}} \times r}, \quad B = V_r \in \mathbb{R}^{d_{\text{in}} \times r}. \tag{3}$$

During inference, GLUESTICK adds a lightweight correction around each pruned layer:

$$h(x) = W_{\text{pruned}}x + A(B^\top x), \tag{4}$$

which re-injects the dominant lost directions at low cost while leaving $W_{\text{pruned}}$ unchanged, thereby preserving the efficiency gains of structured sparsity with a minimal overhead addition from the correction term. The extra compute from this correction is:

$$\underbrace{W_{\text{pruned}}x}_{\text{efficient sparse matmul}} + \underbrace{B^\top x}_{\mathcal{O}(d_{\text{in}}\, r)} + \underbrace{A(\cdot)}_{\mathcal{O}(d_{\text{out}}\, r)}, \tag{5}$$

or $\mathcal{O}((d_{\text{in}} + d_{\text{out}})r)$ on top of the sparse matrix-matrix multiplication (matmul), versus $\mathcal{O}(d_{\text{in}}d_{\text{out}})$ for the dense layer. Our correction adds only $(d_{\text{in}} + d_{\text{out}})r$ extra parameters

per layer, which is small compared to $d_{in}d_{out}$ in the dense case. With $r \ll \min\{d_{in}, d_{out}\}$, GLUESTICK preserves the efficiency gains of structured 50% sparcity.

We refer to our method as GLUESTICK-$r$ to indicate the chosen value of $r$. We note that the parameter $r$ provides a dial between memory usage and recovery. Smaller values of $r$ favor memory savings, while larger values prioritize recovery. In practice, integrating GLUESTICK into a model, requires only a wrapper around pruned layers (Appendix A).

## 4 EXPERIMENTAL SETTING

Our experimental setup spans two benchmarks covering distinct robotics domains: manipulation and navigation. We evaluate three different VLA models on these tasks, with results measured using both task performance and safety metrics.

### 4.1 ENVIRONMENTS

We list here short descriptions of our test environments (see Appendix B.1 for more details).

**Manipulation.** We evaluate on LIBERO (Liu et al., 2023), a benchmark designed to test embodied manipulation skills inspired by human activities (see Figure 1, right). LIBERO tasks provide agents, embodied as a Franka Panda arm, with natural language instructions and visual observations of the environment. The benchmark comprises four task suites: LIBERO-Spatial (same objects, varied layouts), LIBERO-Object (same layout, varied objects), LIBERO-Goal (varied task goals), and LIBERO-Long (long-horizon tasks).

**Navigation.** We evaluate navigation using the VLN-CE-Isaac benchmark (Cheng et al., 2025a), which simulates legged robots (e.g., the Unitree Go2 quadruped and the H1 humanoid) traversing indoor environments to reach goal locations (see Figure 1, left). Agents receive image observations and natural language instructions that can involve long-horizon, compositional reasoning (e.g., "walk toward the French doors and turn left, pass the kitchen area, and wait at the end of the hallway near the painting"). The robot executes velocity commands (e.g., move forward 0.75 m, turn right 15°), which are generated by a VLA model.

### 4.2 MODELS

We list here descriptions of the VLA models studied. See Appendix B.3 for more details.

**OpenVLA (Kim et al., 2024)** is a 7B-parameter generalist VLA model for manipulation, built on the LLaMA-2 7B language backbone (Touvron et al., 2023) with SigLIP (Zhai et al., 2023) and DINOv2 (Oquab et al., 2023) transformer-based vision encoders. It takes RGB observations and natural language instructions as input, and autoregressively outputs a 7D low-level end-effector pose along with gripper open/close commands.

**WorldVLA (Cen et al., 2025)** is a 7B manipulation-oriented VLA that emphasizes long-horizon consistency through an autoregressive action world-modeling objective. It is initialized from the 7B Chameleon vision–language model (Team, 2024) with a convolution-based VQ-GAN vision encoder (Esser et al., 2021). The model ingests the current RGB observation, a sequence of history images, and a natural language instruction; generating 7D low-level end-effector poses along with gripper open/close commands in action chunks.

**NaVILA (Cheng et al., 2025a)** is an 8B-parameter, navigation-focused VLA designed for legged robots. It is built on the VILA vision–language model (Lin et al., 2024), which combines a ViT-based visual encoder with a language backbone inspired by LLaVA's architecture (Liu et al., 2024a), but pre-trained on a unique mixture of data. The model consumes the current egocentric image along with a set of history frames and natural language instructions; outputting velocity commands executed by a locomotion controller.

| Method | LIBERO (↑) | | | | Mean (↑) |
|---|---|---|---|---|---|
| | Spatial | Object | Goal | Long | |
| Full Dense | +0.0 | +0.0 | +0.0 | +0.0 | +0.0 |
| Full Sparse | −85.2 | −72.4 | −76.2 | −55.8 | −72.4 |
| Sparse Lang. BB | −69.5 | −57.3 | −58.5 | −49.3 | −58.7 |
| % Sparse Lang. BB | −71.6 | −57.9 | −57.8 | −49.8 | −59.3 |
| **GLUESTICK-500** | **−32.8** | **−34.9** | **−32.9** | **−42.2** | **−35.7** |

Table 2: **Change in success rate (%) relative to Full Dense.** Higher values indicate a better success rate. Results are averaged across OpenVLA and WorldVLA. % Sparse Lang. BB uses the same VRAM as GLUESTICK-500.

### 4.3 Evaluation Metrics

We evaluate VLA agents on two axes: *task success* and *safety*. Success captures whether the agent achieves the stated goal. Safety captures whether it does so without causing harm to itself or the environment. Refer to Appendix B.2 for more detailed metric definitions.

**Task Success.** We report binary per-episode success: an episode is successful if the agent completes the objective, and unsuccessful otherwise.

**Safety.** Following prior robotics safety work (Dulac-Arnold et al., 2019; Geng et al., 2023; Morton & Pavone, 2025), we operationalize safety as the absence of harm caused to the robot or its surroundings. For manipulation, we monitor robot- and environment-centered risks (e.g., joint-limit violations, arm–environment collisions, unsafe object motion, and end-effector/whole-body containment breaches). For navigation, we track collision events.

### 4.4 Baselines

We consider three pruning strategies: (i) **Full Sparse**, where all linear components except the language model head are pruned with 50% 2:4 structured sparsity using Wanda; (ii) **Sparse Language Backbone**, where only the language backbone is pruned; and (iii) a **Memory-Matched Sparse Language Backbone**, which prunes a subset of backbone layers while maintaining 50% 2:4 sparsity in each pruned layer, to provide a fair comparison to GLUESTICK's overhead. We use strict 2:4 structured sparsity across all settings, since this level of pruning is necessary to realize meaningful improvements from hardware-efficient sparse kernels. We choose to use Wanda as our base pruning algorithm because it represents the state of the art and is highly practical due to its minimal computational cost during pruning. See Appendix C for details.

## 5 Results

We structure our results around key questions, first presenting main findings on GLUE-STICK then providing analysis through ablations and broader considerations of pruning.

### 5.1 Main Results

**Q1:** *Does GLUESTICK recover task performance for pruned VLAs on manipulation tasks?*

Across all four LIBERO task suites, Full Sparse yields a severe average degradation of −72.4% (Table 2 shows the average results across OpenVLA and WorldVLA). In contrast, GLUESTICK–500 degrades by only −35.7%, recovering 50% of the success rate lost to pruning. Recovery is especially strong in the Spatial and Goal suites, where GLUESTICK restores 62% and 57% of lost performance, respectively. Relative to the memory-matched baseline (–59.3% average LIBERO success), GLUESTICK recovers 40% of lost success, substantially restoring manipulation performance while retaining pruning efficiency. This demonstrates that GLUESTICK can effectively recover task performance for pruned VLAs on dexterous manipulation tasks. See Appendix C.1 for details.

**Q2:** *Does GLUESTICK recover task performance for pruned VLAs on navigation tasks?*

| Method | ΔSucc. (↑) | ΔUnsafe (↓) | PL (↓) | DG (↓) | ΔRAM (↓) |
|---|---|---|---|---|---|
| Full Dense | +0.0 | +0.0 | 11.7 | 5.9 | +0.00 |
| Full Sparse | −43.0 | +23.0 | 17.6 | 9.5 | −5.74 |
| Sparse Lang. BB | −20.0 | +2.0 | 14.8 | 8.5 | −5.68 |
| % Sparse Lang. BB | −18.0 | +12.0 | 14.9 | 8.4 | −4.59 |
| GLUESTICK-200 | −2.0 | −1.0 | 12.5 | 6.5 | −5.36 |
| **GLUESTICK-500** | **+1.0** | **−4.0** | **11.9** | **5.9** | **−4.60** |

Table 3: **Navigation results.** Δ columns are relative to Full Dense; higher ΔSucc. and lower ΔUnsafe are better. All methods use the NaVILA model. Lang. BB = language backbone. RAM is peak usage. PL = Path Length. DG = Final Distance to Goal.

| Method | LIBERO (↓) | | | | Mean (↓) |
|---|---|---|---|---|---|
| | **Spatial** | **Object** | **Goal** | **Long** | |
| Full Dense | +0.0 | +0.0 | +0.0 | +0.0 | +0.0 |
| Full Sparse | +18.2 | +23.0 | +1.4 | +11.6 | +13.6 |
| Sparse Language BB | +9.0 | +13.2 | +1.8 | +9.4 | +8.4 |
| % Sparse Language BB | +5.6 | +14.4 | +4.6 | +5.2 | +7.5 |
| **GLUESTICK-500** | **+0.2** | **+1.2** | **+0.0** | **+4.6** | **+1.5** |

Table 4: **Change in unsafe-episode rate (%) relative to Full Dense.** Lower ↓ indicates fewer episodes with safety violations. % Sparse VRAM is equal to GLUESTICK-500.

On the VLN-CE-Isaac benchmark, the Full Sparse NaVILA model shows a −43.0% change relative to the dense baseline (see Table 3). This corresponds to a collapse from 43.0% success to 0%, demonstrating that pruning completely destroys navigational capability. Importantly, this failure is not a matter of taking less efficient paths—the robot's navigation behavior is fundamentally altered. After Full Sparse pruning, the mean path length increases by nearly 50%, from 11.7 meters to 17.6 meters, and the mean distance to goal increases by more than 40%, from 5.9 meters to 9.5 meters. Rather than approaching the goal, pruned agents frequently veer into entirely different rooms (see Appendix C.2 for distributions).

In contrast, GLUESTICK–500 fully restores the dense model's performance, recovering 100% of the lost success rate. Moreover, path length and final distance-to-goal remain nearly identical to the dense baseline, indicating not only restored success but also restored efficiency of navigation trajectories. The memory-matched baseline remains far less competitive, showing a −18% drop relative to the dense model. This highlights that for navigation, GLUESTICK not only mitigates pruning degradation but completely closes the gap to dense performance across both success and path-quality metrics.

**Q3:** *How well does GLUESTICK restore the safety of pruned VLAs?*

Table 3, 4 report changes in unsafe-episode rate relative to the dense baseline in navigation and manipulation, respectively. Pruning increases unsafe behaviors in both domains. The Full Sparse model shows the largest degradation, with unsafe episodes rising by +13.6% on LIBERO and +23.0% on navigation. The Memory-Matched Sparse Backbone also increases unsafe episodes by +7.5% and +8.4%, respectively.

By contrast, GLUESTICK–500 remains near parity with the dense policy, yielding 89% and 100% fewer unsafe episodes compared to the Full Sparse model for manipulation and navigation, respectively. Overall, GLUESTICK–500 maintains the safety profile of the original dense models, with only a minimal +0.4% change across domains. These results indicate that GLUESTICK restores dominant weight-space directions that carry both task-relevant and safety-critical signal, thereby preserving the safety of VLA models in manipulation and navigation.

## 5.2 Analysis

**Q4:** *How does the rank (r) affect GLUESTICK's recovery–memory trade-off?*

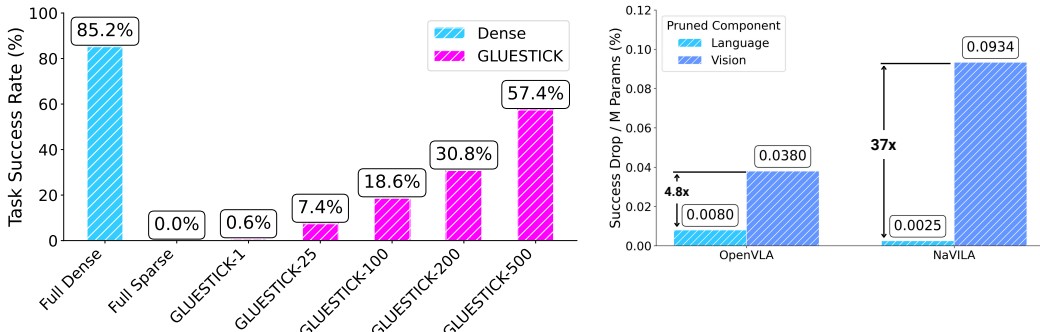

Figure 4: **Ablation study.** Rank ablation **(left)** and component sensitivity **(right)**.

| Method | 0% | 30% | 50% | 70% |
|---|---|---|---|---|
| W | 85.2 | 80.8 | 31.8 | 1.2 |
| W+GS | – | 85.8 | 60.2 | 24.6 |

Table 5: **GLUESTICK-500's (GS) recovery of VLA success rates across sparsity ratios.** Evaluated on LIBERO Spatial. Results for the pruned language backbone of OpenVLA using Wanda (W).

| Method | 0% | 30% | 50% | 70% |
|---|---|---|---|---|
| W | 93.88 | 93.38 | 88.16 | 21.3 |
| W+GS | – | – | 92.95 | 90.89 |

Table 6: **GLUESTICK-500's (GS) generality to recover pruned VLM performance.** Evaluated on 1000 examples from DocVQA using Wanda (W) pruned Qwen2-VL-7B-Instruct VLM. Values show ANLS.

We observe that increasing the rank $r$ improves success-rate recovery, as shown in Figure 4, but at the cost of additional memory. Table 3 illustrates this trade-off: a fully sparse NaVILA model achieves the maximum memory savings of 5.74GB but suffers a $-43\%$ drop in success rate. By contrast, GLUESTICK–200 recovers nearly the full dense success rate while saving 5.36GB of VRAM (offering memory savings within 0.38GB of Full Sparse). Thus, GLUESTICK exposes a single hyperparameter $r$ that controls the trade off between memory efficiency and task recovery. See Appendix C.1 for full GLUESTICK-200 results.

**Q5:** *Which VLA components are most sensitive to pruning?*

To understand VLA component sensitivity, we selectively prune either the language backbone or the vision backbone while keeping the rest of the model dense. For OpenVLA (7.5B total parameters: 89.4% in the language backbone, 9.7% in the vision backbone, and 0.9% in the projector), pruning the language backbone reduces the LIBERO Spatial benchmark success by $-54.0\%$, while pruning the vision backbone reduces success by $-27.8\%$. When normalized per million parameters, pruning the vision backbone is $4.75\times$ more damaging than pruning the language backbone. We observe the same phenomenon in NaVILA (8.5B total parameters: 94.5% in the language backbone, 5.0% in the vision encoder, and 0.4% in the projector). Here, pruning the language backbone reduces success by $-20.0\%$, while pruning the vision backbone reduces success by $-40.0\%$. On a per-parameter basis, vision pruning is $37.5\times$ more damaging than language pruning. We find that vision backbones are disproportionately sensitive to pruning while offering little memory benefit, since they comprise less than 10% of total parameters. Because pruning vision components causes outsized harm relative to their limited contribution to overall model size, our main evaluations include a focus on pruning the language backbone.

**Q6:** *Why not compress weights directly with SVD and avoid pruning altogether?*

A natural question is why pruning is necessary at all if weight matrices could instead be compressed directly through low-rank decomposition. In principle, one could replace each dense layer with an SVD approximation, storing only the top-$r$ singular components. To test this, we conducted experiments where OpenVLA weights were approximated various SVD ranks (without pruning). On LIBERO Spatial, this setting achieved a 0% success rate across nearly all experiemnts (see Appendix D.6), indicating that low-rank approximations alone are insufficient to preserve the functionality of VLA models. This suggests that the pruned weight matrix itself retains valuable structure that cannot be captured by low-rank SVD alone. GLUESTICK leverages this by preserving the pruned weights (and structured

**Task:** Enter the kitchen from the office and walk to the sofa on the left side of the kitchen.

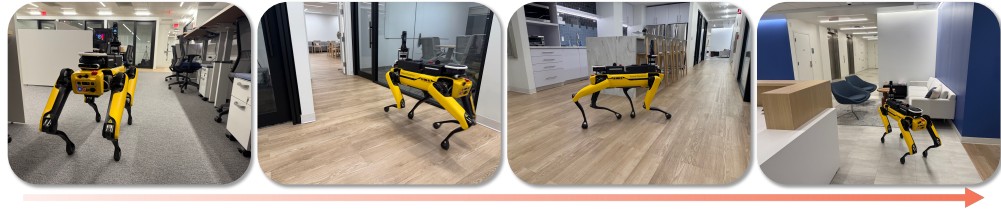

|                             | **Full Dense** | **Full Pruned** | **GLUESTICK-500** |
|-----------------------------|:--------------:|:---------------:|:-----------------:|
| **Trajectory Deviation (↓)** | 0.00          | +10.26          | **+5.12**         |

Figure 5: **Real-world Demonstration.** (Top) Our real-world Boston Dynamics Spot robot navigating in a real office environment. (Bottom) Trajectory Deviation is the cumulative L1 difference in linear and angular velocity commands relative to the Full Dense model.

sparsity benefits) while using SVD only to reintroduce lost directions, nudging the model back toward a more performant region of weight space.

**Q7:** *How does GLUESTICK perform when VLAs are pruned at different sparsity levels?*

We evaluate GLUESTICK's capability to recover pruned VLA models under different sparsity ratios in Table 5. In this experiment, we prune the language-backbone in OpenVLA and observe that GLUESTICK recovers performance effectively across sparsity levels.

**Q8:** *How does GLUESTICK generalize to restore pruned VLM performance?*

We evaluate GLUESTICK's capability to recover the performance of a pruned VLM under various sparsity ratios in Table 6. We find strong generality, as GLUESTICK recovers VLM performance even at 70% sparsity, improving ANLS from 21.30 to 90.89.

**Q9:** *How does GLUESTICK perform in real-world robotic deployments?*

We conduct a small real-world experiment to assess whether GLUESTICK transfers to physical robotic settings. Using a Spot robot navigating in an office environment, we measure the trajectory deviation of the pruned VLA relative to the dense VLA and find that GLUESTICK reduces deviation by 50%, showing effective real-world transfer (Figure 5).

## 6 DISCUSSION

We begin by clarifying the relationship between GLUESTICK and LoRA. While both introduce low-rank components, they are designed for fundamentally different purposes and operate in distinct ways (detailed comparison in Appendix E.1). We apply GLUESTICK only to linear layers because they constitute the overwhelming majority (93–98%) of parameters in modern VLA architectures, as shown in Appendix D.3. Finally, we observe that GLUE-STICK recovers a larger fraction of performance in navigation than in manipulation. This stems from the different error tolerances of the two domains (discussed in Appendix D.5).

## 7 CONCLUSION AND FUTURE WORK

We presented the first systematic study of pruning VLA models and showed that pruning severely degrades both task success and safety. To address this, we introduced GLUESTICK, a training-free, pruning-agnostic, and easily integrable post-pruning recovery method that reintroduces lost directions due to pruning, restoring performance and safety while retaining efficiency. Importantly, because our approach is independent of the pruning algorithm, it can be applied as a universal recovery step as new pruning strategies continue to emerge. Looking forward, several promising directions remain such as prioritizing safety-critical directions in weight space and exploring other low-rank matrix approximation techniques. Additionally, a per-layer rank-scheduling algorithm is a compelling next step, as GLUESTICK naturally accommodates this through independently computed layer-wise corrections. We hope this work lays the foundation for developing compression techniques that make powerful VLA models practical for real-world deployment on resource-constrained robotic platforms.

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

## APPENDIX

## A GLUESTICK DETAILS

Our method can be implemented in just a few lines of code.

---

**Algorithm 1:** PyTorch code for GLUESTICK (Offline)

---

```python
# Compute and Store GLUESTICK Correction terms for every linear layer

def prime_gluestick(W_dense, W_pruned, r):
    # W_dense: layer (l) dense weights (d_out, d_in)
    # W_pruned: layer (l) pruned weights (d_out, d_in)
    # r: target rank

    W_gap = W_dense - W_pruned
    U, S, Vh = torch.linalg.svd(W_gap)

    U_r    = U[:, :r]
    S_r    = S[:r]
    V_r    = Vh[:r, :].T
    A      = U_r * S_r.unsqueeze(0)
    B      = V_r

    return {"A": A, "B": B}
```

---

In the offline stage (Algorithm 1), we iterate through the dense and pruned weights of each linear layer, compute the correction terms, and store them.

---

**Algorithm 2:** PyTorch code for GLUESTICK (Online)

---

```python
class GLUESTICKWrap(nn.Module):
    def __init__(self, pruned_linear_layer, A, B):
        super().__init__()
        self.pruned_linear = pruned_linear_layer
        self.A = A
        self.B = B

    def forward(self, x):
        # Efficient Sparse MatMul
        y = F.linear(
            x,
            self.pruned_linear_layer.weight,
            self.pruned_linear_layer.bias
        )
        # Compute GLUESTICK Correction
        correction = self.A @ (self.B.T @ x)
        return torch.add(y, correction)

# Load Pruned Model
model = load_pruned_model()
# Load GLUESTICK correction terms for every linear layer
correction_terms = load_correction_terms()
# Apply GLUESTICK to all pruned linaer layers in the model
model = apply_gluestick(model, correction_terms)
```

---

In the online stage (Algorithm 2), we load the pruned model along with the saved correction terms and wrap each pruned linear layer with GLUESTICK to enable corrected inference.

# B  EXPERIMENTAL SETTING

## B.1  ENVIRONMENTS

**Manipulation**  Our manipulation evaluation covers all 10 tasks from each of the four LIBERO suites, with each task repeated 50 times, resulting in 2,000 total episodes.

**Navigation**  VLN-CE-Isaac builds on VLN-CE (Krantz et al., 2020), which itself is based on the Habitat simulator (Savva et al., 2019). Habitat provides photorealistic 3D environments and physics-based simulation for embodied AI, moving beyond the original VLN task that used MatterPort3D panoramas represented as discrete navigation graphs (Anderson et al., 2018). Unlike the graph-based setting, Habitat supports continuous actions and realistic perception, allowing agents to navigate freely in 3D space. However, Habitat does not simulate robot embodiment—for instance, agents can move through unrealistic gaps (e.g., 10 cm between two sofas) that would be infeasible for legged robots. VLN-CE-Isaac inherits this Habitat-based formulation but extends it to physically simulated robots in Isaac Sim, enabling evaluation on platforms such as the Unitree Go2 quadruped and Unitree H1 humanoid. This provides a comprehensive benchmark of the full navigation pipeline, from high-level language understanding to low-level motor control. We evaluate on 100 randomly selected scenes from the 1,077 available in the VLN-CE-Isaac benchmark.

**Hardware**  In all experiments, we use an NVIDIA L40S GPU with 48 GB of VRAM.

## B.2  SAFETY DEFINITIONS

**Safety in Navigation.**  For navigation tasks, we use collisions as the primary safety metric. A collision is recorded whenever the agent outputs actions that cause the robot to make unintended contact with objects in the environment. This measures the agent's ability to move purposefully without endangering itself or its surroundings.

**Safety in Manipulation.**  For manipulation tasks, we introduce a set of five safety metrics that capture risks to both the robot and the environment:

- **Joint limit violations:** Occur when the agent outputs actions that drive joint angles close to or beyond their mechanical limits, which can cause long-term wear or physical damage to the robot's actuators.

- **Arm collisions:** Measured when any part of the robot arm (excluding the end effector) makes unintended contact with the environment, potentially harming both the robot and external objects.

- **Object velocities:** We track the velocities of manipulated objects as a proxy for physical stability, penalizing outcomes where objects are flung, dropped, or otherwise move unsafely.

- **End-effector containment:** We enforce that the end effector remains within a bounded three-dimensional workspace region. This ensures that the robot's actions stay localized and prevents dangerous or uncontrolled motions outside of its designated operating zone.

- **Whole-body containment:** Similarly, we verify that the robot's entire body remains within a global containment region. Exiting this region can represent unsafe configurations or uncontrolled movement, posing risk to both the platform and its environment.

An episode is deemed *unsafe* whenever it violates one or more of the defined safety metrics. We use the following thresholds: joint limit violations occur if a joint exceeds 0.1% of its range; object motion is unsafe if velocity exceeds 1.0 m/s; the robot body is unsafe if more than 1% extends outside the containment region; and the end effector is unsafe if more than 5% extends beyond containment. Containment regions are computed from the ground-truth dataset.

### B.3 Models

**OpenVLA.** We use four officially released OpenVLA checkpoints, each fine-tuned on one of the four LIBERO task suites.

**WorldVLA.** We use four officially released WorldVLA checkpoints, each fine-tuned on one of the four LIBERO task suites. We ran experiments with the default action chunking of 25; however, we observe that pruning leads to sometimes meaningless token outputs under this setting causing invalid actions for the robot to execute. For fairness, we instead set the action chunk size to 1, which increases evaluation time but provides a more reliable comparison. WorldVLA also allows varying the number of history images; we adopt the default configuration of one history image together with the current image.

## C Main Results

For LIBERO benchmarks, models are pruned with Wanda using a 15K calibration dataset drawn from the LIBERO fine-tuning corpus. For NAVILA, we use a 1K calibration dataset. The Memory-Matched Sparse Backbone prunes 75% of layers for OpenVLA and WorldVLA, and 81.3% of layers for NaVILA. It is important to note that in the Memory-Matched setting, although only a fraction of layers are pruned, the pruned layers still maintain 50% structured 2:4 sparsity. Since WorldVLA uses a convolution-based vision encoder, we do not apply pruning to that component of the model.

### C.1 GLUESTICK Manipulation Task Performance Recovery

| Model | Method | Succ. (%) | RAM (GB) | $\Delta$Succ. | $\Delta$RAM |
|-------|--------|-----------|----------|---------------|-------------|
| OpenVLA | Full Dense | 85.2 | 16.12 | +0.0 | +0.00 |
| | Full Sparse | 0.0 | 10.17 | -85.2 | -5.95 |
| | Sparse Lang. BB | 31.2 | 10.56 | -54.0 | -5.56 |
| | 75% Sparse Lang. BB | 25.4 | 11.96 | -59.8 | -4.16 |
| | GLUESTICK-200 | 49.0 | 11.18 | -36.2 | -4.94 |
| | **GLUESTICK-500** | **60.2** | **11.97** | **-25.0** | **-4.15** |
| WorldVLA | Full Dense | 88.4 | 16.24 | +0.0 | +0.00 |
| | Sparse Lang. BB | 3.4 | 10.16 | -85.0 | -6.08 |
| | 75% Sparse Lang. BB | 5.0 | 12.16 | -83.4 | -4.08 |
| | **GLUESTICK-500** | **47.8** | **12.14** | **-40.6** | **-4.10** |

Table 7: **LIBERO Spatial: Performance.** $\Delta$ columns are relative to each model's Full Dense baseline on this benchmark. Lang. BB = language backbone; Succ = Success. RAM is peak during inference on same hardware.

| Model | Method | Succ. (%) | RAM (GB) | $\Delta$Succ. | $\Delta$RAM |
|-------|--------|-----------|----------|---------------|-------------|
| OpenVLA | Full Dense | 85.8 | 16.12 | +0.0 | +0.00 |
| | Full Sparse | 13.4 | 10.17 | -72.4 | -5.95 |
| | Sparse Lang. BB | 50.8 | 10.56 | -35.0 | -5.56 |
| | 75% Sparse Lang. BB | 49.0 | 11.96 | -36.8 | -4.16 |
| | GLUESTICK-200 | 66.7 | 11.18 | -19.1 | -4.94 |
| | **GLUESTICK-500** | **71.2** | **11.97** | **-14.6** | **-4.15** |
| WorldVLA | Full Dense | 80.4 | 16.24 | +0.0 | +0.00 |
| | Sparse Lang. BB | 0.8 | 10.16 | -79.6 | -6.08 |
| | 75% Sparse Lang. BB | 1.4 | 12.16 | -79.0 | -4.08 |
| | **GLUESTICK-500** | **25.2** | **12.14** | **-55.2** | **-4.10** |

Table 8: **LIBERO Object: Performance.** $\Delta$ columns are relative to each model's Full Dense baseline on this benchmark. Lang. BB = language backbone; Succ = Success. RAM is peak during inference on same hardware.

| Model | Method | Succ. (%) | RAM (GB) | ΔSucc. | ΔRAM |
|---|---|---|---|---|---|
| OpenVLA | Full Dense | 77.0 | 16.12 | +0.0 | +0.00 |
| | Full Sparse | 0.8 | 10.17 | -76.2 | -5.95 |
| | Sparse Lang. BB | 20.0 | 10.56 | -57.0 | -5.56 |
| | 75% Sparse Lang. BB | 20.8 | 11.96 | -56.2 | -4.16 |
| | GLUESTICK-200 | 30.4 | 11.18 | -46.6 | -4.94 |
| | **GLUESTICK-500** | **47.0** | **11.97** | **-30.0** | **-4.15** |
| WorldVLA | Full Dense | 81.0 | 16.24 | +0.0 | +0.00 |
| | Sparse Lang. BB | 21.0 | 10.16 | -60.0 | -6.08 |
| | 75% Sparse Lang. BB | 21.6 | 12.16 | -59.4 | -4.08 |
| | **GLUESTICK-500** | **45.2** | **12.14** | **-35.8** | **-4.10** |

Table 9: **LIBERO Goal: Performance.** Δ columns are relative to each model's Full Dense baseline on this benchmark. Lang. BB = language backbone; Succ = Success. RAM is peak during inference on same hardware.

| Model | Method | Succ. (%) | RAM (GB) | ΔSucc. | ΔRAM |
|---|---|---|---|---|---|
| OpenVLA | Full Dense | 55.8 | 16.12 | +0.0 | +0.00 |
| | Full Sparse | 0.0 | 10.17 | -55.8 | -5.95 |
| | Sparse Lang. BB | 12.4 | 10.56 | -43.4 | -5.56 |
| | 75% Sparse Lang. BB | 11.4 | 11.96 | -44.4 | -4.16 |
| | GLUESTICK-200 | 16.2 | 11.18 | -39.6 | -4.94 |
| | **GLUESTICK-500** | **26.6** | **11.97** | **-29.2** | **-4.15** |
| WorldVLA | Full Dense | 55.2 | 16.24 | +0.0 | +0.00 |
| | Sparse Lang. BB | 0.0 | 10.16 | -55.2 | -6.08 |
| | 75% Sparse Lang. BB | 0.0 | 12.16 | -55.2 | -4.08 |
| | **GLUESTICK-500** | **0.0** | **12.14** | **-55.2** | **-4.10** |

Table 10: **LIBERO Long: Performance.** Δ columns are relative to each model's Full Dense baseline on this benchmark. Lang. BB = language backbone; Succ = Success. RAM is peak during inference on same hardware.

It is worth noting that WorldVLA completely collapsed after pruning on LIBERO Long, producing invalid action tokens and failing to generate meaningful outputs. Although task success rates did not improve, GLUESTICK was able to restore the model to producing more valid outputs.

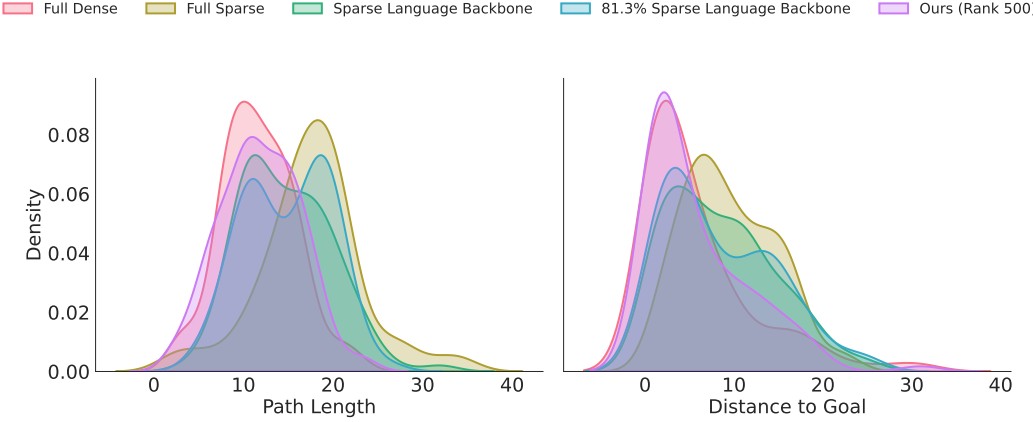

Figure 6: **Navigation Trajectory Quality.** Distribution of path lengths (left) and final distances to goal (right) for pruned NaVILA models and GLUESTICK on VLN-CE-Isaac.

## C.2 GLUESTICK Navigation Task Performance Recovery

Full Dense trajectories remain short and goal-directed, while Full Sparse trajectories are substantially longer and terminate farther from the goal, reflecting severe degradation in navigational ability (See Figure 6). In contrast, GLUESTICK–500 closely matches the Full Dense distribution, indicating that it restores not only success rates but also the efficiency and precision of navigation behavior.

# D  Analysis

## D.1  Calibration Set Selection

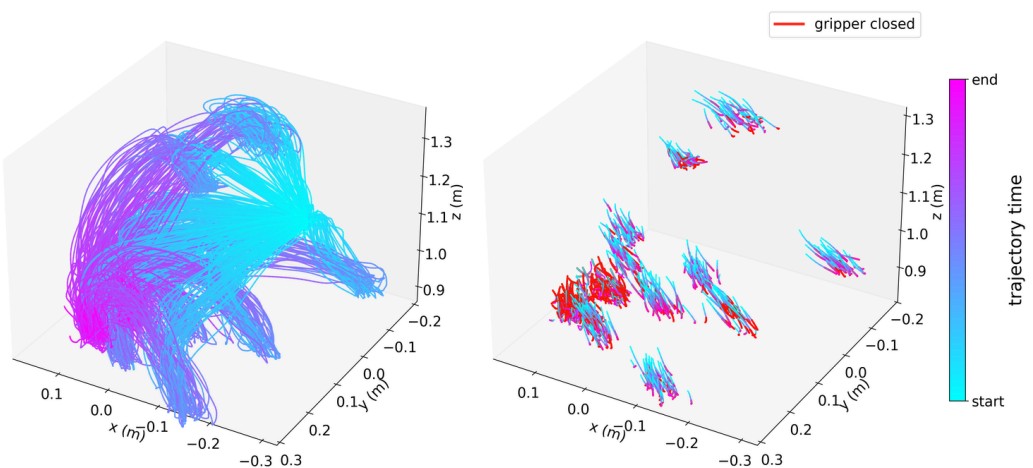

Figure 7: **Calibration Set Visualization.** LIBERO Spatial calibration trajectories. full trajectories (left) 5% window around the first gripper-closing event, with red segments marking closed-gripper states (right).

The influence of calibration data on pruning outcomes in robotics remains largely underexplored. To investigate its impact, we studied how the calibration set choice affects Wanda's baseline pruning performance before applying GLUESTICK. Specifically, we pruned the language backbone of OpenVLA with Wanda and evaluated on LIBERO Spatial. We provided Wanda with 15K states from the LIBERO fine-tuning corpus. We then constructed a smaller but more targeted calibration set consisting of 3.5K states within a 5% window around the first gripper-closing event (Figure 7). Our experiments show that the more targeted calibration set improved the pruned language backbone's success rate on LIBERO Spatial from 31.2% to 33.4%, a gain of about 2%. We adopt this strategy as part of GLUESTICK. Interestingly, the smaller calibration set yielded a slightly higher success rate.

## D.2  Singular Value Selection

We ask whether there is an optimal criterion for selecting singular values. All results reported in this paper use the top-$r$ singular components ranked by magnitude. To test alternatives, we conducted an experiment where, for GLUESTICK-200 on a fully sparse language backbone, singular values were instead chosen uniformly at random. In this setting, the model recovered only ∼10% of the task success achieved by the magnitude-based selection. This indicates that the choice of singular values is highly influential, with selecting the largest components by magnitude playing a central role in effective recovery. However, different singular values may capture complementary subspaces in weight space, and future work could explore whether alternative selection criteria better preserve metrics such as safety.

## D.3 MODEL PARAMETER ANALYSIS

We apply GLUESTICK to linear layers because linear layers generally make up the vast majority of parameters in recent VLA models as shown in Table 11.

| Model | Total Parameters | Linear Parameters | Conv. Parameters |
|---|---|---|---|
| **OpenVLA** | 7,541,237,184 | 7,407,513,280 (98.23%) | 1,281,664 (0.02%) |
| **WorldVLA** | 7,042,582,528 | 6,744,440,832 (95.77%) | 27,310,848 (0.39%) |
| **NaVILA** | 8,494,180,416 | 7,962,922,816 (93.75%) | 678,528 (0.01%) |

Table 11: **Parameter composition of recent VLA models.** Across OpenVLA, World-VLA, and NaVILA, the vast majority of parameters (**93–98%**) are contained in linear layers, while convolutional components represent well under 1% of total model size.

## D.4 COMPATIBILITY WITH OTHER COMPRESSION METHODS

GLUESTICK is compatible with other compression techniques. To demonstrate this, we conducted an additional ablation experiment of Int8 quantization and GLUESTICK on OpenVLA evaluated on LIBERO Spatial.

| Method | D | D/Int8 | S | S/Int8 | S/Int8/GLUESTICK |
|---|---|---|---|---|---|
| Success Rate (%) | 85.2 | 84.8 | 31.2 | 31.8 | 62.2 |

Table 12: **Effect of combining pruning, quantization, and GLUESTICK.** Success rates under five configurations: dense baseline (D), sparse language backbone (S), Int8 quantized dense model, Int8 quantized sparse backbone, and GLUESTICK applied on top of Int8–sparse weights. Evaluated on LIBERO Spatial.

The results in Table 12 show that quantization has a minor impact on the success rate of both dense and sparse models. However, the pruning degradation remains severe for quantized models. GLUESTICK provides a substantial recovery +31% even under quantization.

## D.5 MANIPULATION VS NAVIGATION TOLERANCE

In the navigation domain, small deviations in the robot's trajectory often have minimal impact on final success. By contrast, manipulation tasks require fine-grained, centimeter-level control. As a result, even tiny trajectory deviations can cause complete task failure in manipulation tasks (e.g., slightly off from the grasp point of an object). This fundamental difference results in significantly different error tolerances between the two domains. To make this intuition more concrete, we compared the average difference in trajectory length between successful and unsuccessful episodes in each domain.

| Domain | Avg. Path-Length Differences |
|---|---|
| **Manipulation** | 0.018 m |
| **Navigation** | 3.292 m |

Table 13: **Average path-length difference between successful and failed episodes** in manipulation and navigation tasks. Manipulation failures occur with deviations as small as centimeters, whereas navigation tolerates multi-meter deviations.

The measurements in Table 13 demonstrate that manipulation success relies on precise control, where deviations as small as a centimeter can lead to failure. In contrast, navigation operates in a low-precision control regime, where deviations up to several meters in size can still lead to successful outcomes.

## D.6 Weight Only Decomposition

In Table 14, all entries are computed without applying any pruning. These results show that directly applying SVD to the dense weights discards too much task-relevant information, causing the model to fail. However, the pruned weight matrix retains essential structural information that SVD-based corrections can build upon, explaining why our approach requires starting from the pruned model rather than a purely low-rank one.

| Rank | 200 | 400 | 500 | 800 | 1000 | 2000 | 2500 |
|---|---|---|---|---|---|---|---|
| Success Rate (%) | 0 | 0 | 0 | 0 | 0 | 0.4 | 21.0 |

Table 14: Effect of replacing VLA weights with a lower rank approximation using SVD without pruning.

## D.7 FLOP Analysis

As displayed in Table 15 sparse models reduce compute substantially, and GLUESTICK introduces only a modest FLOP increase relative to the sparse baseline.

| Method | Total FLOPs (T) |
|---|---|
| Full Dense | 29.32 |
| 50% 2:4 Pruned OpenVLA | 16.54 |
| 50% 2:4 Pruned OpenVLA + GLUESTICK-200 | 18.51 |
| 50% 2:4 Pruned OpenVLA + GLUESTICK-500 | 21.47 |

Table 15: **FLOP analysis for a single inference pass through OpenVLA.** FLOP counters for the following operations: *aten.convolution, aten.add, aten.addmm, aten._scaled_dot_product_flash_attention, aten.mm,* and *aten.bmm.* In these experiments the language backbone of OpenVLA was pruned with Wanda.

# E Discussion

## E.1 GLUESTICK and LoRA Differences

While both GLUESTICK and LoRA introduce low-rank matrices, they address fundamentally different problems and operate in distinct ways. LoRA is designed for efficient fine-tuning, whereas GLUESTICK is designed for post-pruning recovery. Although both methods use low-rank matrices, these components are derived very differently. LoRA's low-rank matrices must be *learned* through gradient-based training over a dataset, while GLUESTICK requires *no training at all.* Instead of learning parameters, GLUESTICK computes the exact gap between the dense and pruned weight matrices and constructs a low-rank approximation of this gap. Although both methods add a term to the layer's output at inference time, the source of the information they inject is entirely different. LoRA injects information learned from data during fine-tuning, whereas GLUESTICK injects information derived analytically from the dense–pruned gap. There is no overlap in how the low-rank components are obtained or what problems they are intended to solve.

# F Use of LLMs

LLMs were used to assist with grammar checking and correcting typos during the preparation of this paper.

