# OpenReview forum: "Don't Run with Scissors: Pruning Breaks VLA Models but They Can Be Recovered"
_ICLR.cc/2026/Conference — Submitted to ICLR 2026_

### Official Review · Reviewer_aTbD · 2025-10-28

**Soundness:** 4
**Presentation:** 4
**Contribution:** 3
**Rating:** 10
**Confidence:** 4

**Summary:**

This paper is about the very important question of how to make a VLA model relevant (fast) by pruning without deteriorating the performance too much. The approach computes a correction term that is based on the un-pruned and the pruned model that is later used in inference with the pruned model. The term is only computed once and required not knowledge of the pruning method. The paper provides empirical evidence of the problem actually existing, provides insights as to why the problem arises, and demonstrates that the proposed method solved the issue.

**Strengths:**

- The paper addresses a relevant and important problem.

- The paper presents a good explanation and demonstration that the issue exists and is relevant.

- The paper presents an effective solution to the problem.

- The approach has favorable properties such as only computing the correction term once and being independent of the pruning method.

- The introduction and related work sections are well formulated.

- The method is innovative, making use of a low-dimensional correction term (from SVD).

- The paper contains code examples for the most important parts of the approach.

**Weaknesses:**

- The presentation of figure 2 is hard to read and some different way of presenting the same information would help.

**Questions:**

- The proposed approach computes the correction term after pruning and which happens after learning the model in the first place. Would it be possible to improve the correction performance by pruning in a certain way or learning parameters that make correction with the SVD approach easier?

- What bias is the SVD approach introducing to the correction?

- There exist other low-rank decompositions of matrices. Why is the SVD one preferred?

---

> ### Author Response · Authors · 2025-11-28
> **Response by Authors [Part 1]**
>
> We sincerely thank the reviewer for the thoughtful and encouraging assessment of our work. We deeply appreciate that the reviewer strongly supported the importance of recovering pruned VLA models, given our findings of how severely SOTA pruning methods damage them. We are grateful that the reviewer highlighted the strengths of our empirical analysis, the clarity of our explanation of why pruning failures occur, and the effectiveness of our proposed solution. We also appreciate the reviewer's recognition of the novelty and usefulness of our low-dimensional corrective term, and we are encouraged by the positive feedback on the soundness, presentation, and contribution of the work. We appreciate the reviewer for raising insightful questions, which we address in detail below.
>
> ---
>
> > **[W1]** The presentation of figure 2 is hard to read and some different way of presenting the same information would help.
>
> We thank the reviewer for this helpful feedback on how to better communicate our findings about the structural differences between VLA and LLM weight spaces through our spectral analysis. We have improved the figure's formatting, added another model comparison, and added a more detailed explanation in Section 3.1 of our revised paper. We hope these adjustments convey our findings more clearly, namely that VLA models exhibit a noticeably flatter spectral decay, with energy spread across many directions, which makes them more vulnerable to pruning.
>
> ---
>
> > **[Q1]** The proposed approach computes the correction term after pruning and which happens after learning the model in the first place. Would it be possible to improve the correction performance by pruning in a certain way or learning parameters that make correction with the SVD approach easier?
>
> As the reviewer pointed out, our method is training-free, which offers strong practical advantages: GLUESTICK can be applied post-hoc to any pruned model with negligible computational overhead. At the same time, we agree with the reviewer that there is exciting room to further improve the correction itself while still avoiding any additional training. Our spectral analyses show that VLA layers exhibit much flatter singular value spectra than LLMs, meaning important signal is dispersed across many directions. GLUESTICK addresses this by reinstating the top-$r$ dominant lost directions, but we envision that even richer selection strategies could enhance performance. In particular, we hypothesize that different singular directions correspond to distinct functional subspaces in weight space, for example directions contributing to smoother trajectories, collision-avoidance tendencies, or multi-step reasoning. This intuition suggests a natural extension: instead of selecting components purely by magnitude, one can choose correction directions based on what behaviors they help recover. Such attribute-aware subspace selection would allow practitioners to reinforce specific capabilities (e.g., safety-critical behaviors) while still preserving the simplicity and training-free nature of GLUESTICK.
>
> We appreciate the reviewer's encouragement and view this as an especially promising and exciting direction for future refinement of our method. We have revised our Future Work section (Section 7) to reflect this suggestion.
>
> ---
>
> > **[Q2]** What bias is the SVD approach introducing to the correction?
>
> We can view the bias introduced by our method as the SVD-based corrections that nudge the output of each layer. This biases the correction toward the dominant lost directions in weight space, meaning the directions that account for the largest portion of information removed during pruning. These dominant components are not externally imposed; they are derived entirely from the model’s own lost weights. As a result, the correction does not introduce new data, assumptions, or behavior. Rather, it selectively restores the portions of the original model that pruning removed. In other words, the SVD provides a structured way of re-injecting informative lost directions while discarding low-energy components that contribute minimally to the dense model.

---

> ### Author Response · Authors · 2025-11-28
> **Response by Authors [Part 2]**
>
> > **[Q3]** There exist other low-rank decompositions of matrices. Why is the SVD one preferred?
>
> The reviewer raises an excellent point regarding alternative low-rank decompositions. We choose the SVD because it has several properties that are unique and especially useful in our setting. First, SVD provides a theoretically guaranteed optimal low-rank approximation of a matrix. As a result, truncating SVD at rank $r$, $r$ gives a provably best reconstruction, which is important when trying to isolate the directions of greatest importance before pruning. Second, the singular values calculated by SVD offer a clear and interpretable criterion for rank selection. Because singular values are ordered by magnitude, they directly indicate how much information each component contributes. This makes per-layer rank selection easily analyzable. We agree with the reviewer that other low-rank matrix approximation techniques exist. This opens a very interesting direction for future work, where exploring alternative matrix decomposition methods may provide further gains for GLUESTICK.  We have updated the Future Work section (Section 7) of our revised paper to reflect this insightful suggestion.

---

### Official Review · Reviewer_sjPP · 2025-10-30

**Soundness:** 3
**Presentation:** 3
**Contribution:** 3
**Rating:** 6
**Confidence:** 3

**Summary:**

This paper introduces GLUESTICK, a training-free, pruning-agnostic post-pruning recovery method. While pruning is an effective compression technique for LLMs, it causes drastic degradation in VLA models, leading to near-zero task success rates and increased safety violations. GLUESTICK computes a lightweight corrective term via SVD of the difference between dense and pruned weights and then applies this correction during inference.  A single interpretable hyperparameter rank $r$ is used to balance efficiency and accuracy. The overall method is simple and easy to implement. Experimental results across several VLA models and benchmarks show that GLUESTICK can help recover most of the lost performance while maintaining memory efficiency.

**Strengths:**

- The paper’s observation of the pruning collapse issue in Vision-Language-Action (VLA) models is quite meaningful.
- The proposed method is simple, efficient, and easy to implement, compatible with various pruning techniques.
- It offers valuable insights for the compression, pruning, and deployment of VLA models.

**Weaknesses:**

- The method presented in this paper bears some similarity to adding a low-rank adapter on top of pruning to offset pruning-induced losses. It would be better for the authors to elaborate on the differences between their proposed method and approaches like LoRA.
- When utilizing different backbones and dimensions, how should the hyperparameter $r$ (rank) be determined for each scenario? Would it be feasible to assign distinct $r$ values to different weight matrices? This adjustment seems promising for further enhancing the trade-off between performance and efficiency.
- The performance of the method under different pruning sparsity levels is not explored.

**Questions:**

- Does the proposed method have any impact on inference speed and latency?
- In practice, VLA deployment often requires combining pruning with other methods like quantization. Does GLUESTICK still work with these techniques?

---

> ### Author Response · Authors · 2025-11-28
> **Response by Authors [Part 1]**
>
> Thank you for your valuable time and helpful feedback in improving our paper. We are glad to hear that you found our proposed method “simple, efficient, and easy to implement, compatible with various pruning techniques”. We are also glad to hear that you found the collapse of both safety and performance when pruning VLAs to be “quite meaningful”, and that this insight offers value to the community for their “compression, pruning, and deployment of VLA models”. We provide detailed responses and additional experiments to address the reviewer's questions and comments below.
>
> ---
>
> > **[W1]** ... It would be better for the authors to elaborate on the differences between their proposed method and approaches like LoRA.
>
> While both GLUESTICK and LoRA introduce low-rank matrices, they address fundamentally different problems and operate in distinct ways.
>
> * **(1)** LoRA is designed for efficient fine-tuning, whereas GLUESTICK is designed for post-pruning recovery.
>
> * **(2)** Although both methods use low-rank matrices, these components are derived very differently. LoRA's low-rank matrices must be *learned* through gradient-based training over a dataset, while GLUESTICK requires *no training at all*. Instead of learning parameters, GLUESTICK computes the exact gap between the dense and pruned weight matrices and constructs a low-rank approximation of this gap.
>
> * **(3)** Although both methods add a term to the layer’s output at inference time, the source of the information they inject is entirely different. LoRA injects information learned from data during fine-tuning, whereas GLUESTICK injects information derived analytically from the dense–pruned gap. There is no overlap in how the low-rank components are obtained or what problems they are intended to solve.
>
> We thank the reviewer for raising this point, as it highlights an important distinction and the novelty of our approach. We have added a clarification in the paper describing these differences in our Discussion section (Section 6) and Appendix E.1.
>
> ---
>
> > **[W3]** The performance of the method under different pruning sparsity levels is not explored.
>
> We conducted additional experiments across other sparsity levels. The table below reports the impact of pruning the language backbone of OpenVLA using Wanda on the LIBERO Spatial benchmark. These results show that GLUESTICK improves performance across sparsities. However, our paper’s focus remains on 50% 2:4 structured sparsity, which is most useful in robotics due to CUDA kernel support. These new results are added in Table 5 and Section 5.2 of our revised main paper.
>
> |Base Model| **Method**                 | **0%** | **30%** | **50%** | **70%** |
> |-|---------------------------|--------|---------|---------|---------|
> |OpenVLA (VLA)| Wanda                 | 85.2   | 80.8    | 31.8    | 1.2     |
> |OpenVLA (VLA)| Wanda + GLUESTICK-500 | No Need      | 85.8    | 60.2    | 24.6    |   |
>
> ---
>
> > **[Q1]** Does the proposed method have any impact on inference speed and latency?
>
> GLUESTICK was designed to preserve much of the efficiency benefits of structured sparsity. Because GLUESTICK adds only lightweight low-rank corrections, its overhead is intentionally small. To quantify this, we performed a standard FLOP analysis. Specifically, this analysis is done by tracking the FLOPs spent by key operations over a single inference of the OpenVLA model. We measured FLOPs across the following core operations: *aten.convolution*, *aten.add*, *aten.addmm*, *aten._scaled_dot_product_flash_attention*, *aten.mm*, and *aten.bmm*.
>
> | **Method**                        | **Total FLOPs (T)** |
> |----------------------------------|----------------------|
> | Full Dense OpenVLA                 | 29.32               |
> | 50% 2:4 Pruned OpenVLA          | 16.54               |
> | 50% 2:4 Pruned OpenVLA + GLUESTICK-200 | 18.51           |
> | 50% 2:4 Pruned OpenVLA + GLUESTICK-500 | 21.47           |
>
> *(Lower FLOPs is better)*
>
> From this experiment, we can see that GLUESTICK's choice of $r$ provides a dial that can allow practitioners to trade-off latency in addition to memory. In the case of GLUESTICK-200, GLUESTICK preserves most of the sparse model’s FLOP savings (85%). This new analysis is added in Appendix D.7.

---

> ### Author Response · Authors · 2025-11-28
> **Response by Authors [Part 2]**
>
> > **[Q2]** In practice, VLA deployment often requires combining pruning with other methods like quantization. Does GLUESTICK still work with these techniques?
>
> GLUESTICK is fully compatible with other compression techniques. To demonstrate this, we conducted an additional ablation experiment of Int8 quantization and GLUESTICK. Our experimentation in the table below evaluates the OpenVLA model on the LIBERO Spatial benchmark.
>
> | **Method** | **Dense** | **Dense & Int8** | **Sparse** | **Sparse & Int8** | **Sparse & Int8 & GLUESTICK** |
> |------------|-------|------------|-------|------------|----------------------|
> | Success Rate (%) | 85.2 | 84.8 | 31.2 | 31.8 | 62.2 |
>
> These results show that quantization has a minor impact on the success rate of both dense and sparse models. However, the pruning degradation remains severe for quantized models. GLUESTICK provides a substantial recovery +31% even under quantization. This result supports our claim that GLUESTICK can be applied with other compression techniques. We have updated Appendix D.4 to include this exciting result.
>
> ---
>
> > **[W2]** When utilizing different backbones and dimensions, how should the hyperparameter
>  (rank) be determined for each scenario? Would it be feasible to assign distinct
>  values to different weight matrices? This adjustment seems promising for further enhancing the trade-off between performance and efficiency.
>
> In our experiments, we used a single global rank across all backbones and dimensions, and we found that even this simple setting yielded strong performance recovery. As the reviewer suggests, assigning different ranks to different weight matrices is certainly feasible. GLUESTICK naturally supports this because it computes its correction for each layer independently, making per-layer rank selection straightforward to incorporate. This is an exciting direction for further improving the performance–efficiency trade-off, and we have updated the future work section (Section 7) of our revised paper to reflect this insightful suggestion.

---

### Official Review · Reviewer_yHH6 · 2025-11-01

**Soundness:** 3
**Presentation:** 2
**Contribution:** 3
**Rating:** 4
**Confidence:** 3

**Summary:**

The paper starts from a observation from standard, LLM-validated pruning catastrophically collapses VLA policies—success drops to 0% on both manipulation (OpenVLA: 85.2% to 0%) and navigation (NaVILA: 43% to 0%). It then proposes GLUESTICK, a training-free, pruning-agnostic weight-space correction: compute the dense–pruned gap per linear layer, take a truncated SVD, and add a low-rank correction at inference to restore “lost directions”. GLUESTICK substantially recovers manipulation (≈50% of success lost to pruning) and fully restores navigation success while keeping most of the VRAM savings of structured sparsity; unsafe-episode rates return near dense baselines. The paper further diagnoses why VLAs are fragile: compared to LLM layers, VLA layers show flatter singular spectra, meaning “useful signal” is spread across many directions and is easily excised by structured pruning.

**Strengths:**

* Clear empirical finding: the results are quite good across two domains (manipulation/navigation) and three architectures.
* Simple method arch: GLUESTICK is training-free, drop-in, and pruning-agnostic; a single interpretable hyperparameter r controls the memory-recovery trade-off.
* Thoughtful analysis: the analysis provides a plausible reason VLAs differ from LLMs (flatter spectra -> pruning removes distributed, important directions), which aligns with the effectiveness of a low-rank “stitch-back” on top of pruned weights.

**Weaknesses:**

* Corrections applied only to the linear layers: for the model with heavy conv layers for vision encoders or attention projection with structured kernels, the proposed method might lose some effectiveness, like the WorldVLA case.
* The rank scheduling is empirical: with a single global r used all-way, given large per-layer variation, and the vision backbone is sensitive parameter-wise, without considering the inner difference of different layers.
* The requirement of dense-weights: the method needs the original dense checkpoint to compute the gap SVD, which might constrain its usage in some scenarios.

**Questions:**

* Is it possible for the method to interplay with other techniques like quantization or LoRA? Could you try with more compression baselines for more comprehensive ablation studies?
* You have mentioned that the manipulation task can only achieve ~50% of recovery; can you analyze more on the performance difference with different task settings?
* How sensitive is the method to domain shift and long-horizon tasks?
* In Appendix D.1, the authors find that a smaller, more "targeted" calibration set for Wanda pruning yields a 2% performance gain. This is an interesting but counter-intuitive result. Does this suggest that pruning methods are highly sensitive to the quality and relevance of calibration data, and that "more data" is not always better?

---

> ### Author Response · Authors · 2025-11-28
> **Response by Authors [Part 1]**
>
> We thank the reviewer for recognizing our strong experimental results, the benefits of a simple method architecture, and a thoughtful analysis of why VLAs differ from LLMs.
>
> We greatly appreciate the reviewer's insights and questions. Below, we address each comment, and we have carefully conducted all additional experiments requested based on this reviewer's thoughtful feedback. We have also incorporated the reviewer's suggestions into an updated version of the main paper and appendix.
>
> ---
>
> > **[W1]** Corrections applied only to the linear layers: for the model with heavy conv layers for vision encoders or attention projection with structured kernels, the proposed method might lose some effectiveness, like the WorldVLA case.
>
> We agree with the reviewer's observation that our method currently applies corrections only to linear layers. However, please note that we apply GLUESTICK only to linear layers because linear layers generally make up the vast majority of parameters in recent VLA models (93–98%), as shown below:
>
> | **Model**   | **Total Parameters**     | **Linear Parameters**                  | **Conv. Parameters**                 |
> |-------------|---------------------------|----------------------------------------|--------------------------------------|
> | OpenVLA | 7,541,237,184             | 7,407,513,280 (98.23%)                 | 1,281,664 **(0.02%)**                    |
> | WorldVLA| 7,042,582,528             | 6,744,440,832 (95.77%)                 | 27,310,848 **(0.39%)**                   |
> | NaVILA  | 8,494,180,416             | 7,962,922,816 (93.75%)                 | 678,528 **(0.01%)**
>
> While pruning additional types of layers offers the opportunity for further improvements in memory, their impact will be marginal in current VLA implementations. Therefore, we choose to focus on linear layers in this work. Details are included in our revised paper (Section 6 and Appendix D.3).
>
> ---
>
> > **[Q1]** Is it possible for the method to interplay with other techniques like quantization or LoRA? Could you try with more compression baselines for more comprehensive ablation studies?
>
> GLUESTICK is fully compatible with other compression techniques. To demonstrate this, we conducted an additional ablation experiment of Int8 quantization and GLUESTICK. Our experimentation in the table below evaluates the OpenVLA model on the LIBERO Spatial benchmark.
>
> | **Method** | **Dense** | **Dense & Int8** | **Sparse** | **Sparse & Int8** | **Sparse & Int8 & GLUESTICK** |
> |------------|-------|------------|-------|------------|----------------------|
> | Success Rate (%) | 85.2 | 84.8 | 31.2 | 31.8 | 62.2 |
>
> These results show that quantization has a minor impact on the success rate of both dense and sparse models. However, the pruning degradation remains severe for quantized models. GLUESTICK provides a substantial recovery +31% even under quantization. This result supports our claim that GLUESTICK can be applied with other compression techniques. We have updated Appendix D.4 to include this exciting result.
>
> ---
>
> > **[Q2]** You have mentioned that the manipulation task can only achieve ~50% of recovery; can you analyze more on the performance difference with different task settings?
>
> In the navigation domain, small deviations in the robot's trajectory often have minimal impact on final success. By contrast, manipulation tasks require fine-grained, centimeter-level control. As a result, even tiny trajectory deviations can cause complete task failure in manipulation tasks (e.g., slightly off from the grasp point of an object). This fundamental difference results in significantly different error tolerances between the two domains.
> To make this intuition more concrete, we compared the average difference in trajectory length between successful and unsuccessful episodes in each domain.
>
> | **Domain**         | **Average Path-Length Difference Between Success and Failure Cases** |
> |--------------------|---------------------------------------------------------------|
> | Manipulation   | 0.018 m                                                       |
> | Navigation     | 3.292 m                                                       |
>
> These measurements show that manipulation success relies on precise control, where deviations as small as a centimeter can lead to failure. In contrast, navigation operates in a low-precision control regime, where deviations up to several meters in size can still lead to successful outcomes. We have added this intuition to our revised paper's discussion section (Section 6) and Appendix D.5.

---

> ### Author Response · Authors · 2025-11-28
> **Response by Authors [Part 2]**
>
> > **[Q3]** How sensitive is the method to domain shift and long-horizon tasks?
>
> We can understand the applicability of GLUESTICK to long-horizon tasks based on our extensive LIBERO-Long and Navigation (VLN-CE-Isaac) results. We also conduct new experimentation to test GLUESTICK’s applicability to domain shift.
>
> **(1) Description of our long-horizon experimentation:** LIBERO-Long’s tasks were designed to include a composition of multiple subtasks. Additionally, our navigation evaluations are long-horizon in nature, as their instructions also involve multiple subtasks.
>
> For example, consider the instructions from the three settings below. LIBERO-Long and Navigation (VLN-CE-Isaac) contain two or more tasks to complete, reflecting the long-horizon nature of these tasks:
>
> * LIBERO-Spatial: ```place the black bowl on the plate```
>
> * LIBERO-Long: ```turn on the stove and put the moka pot on it```
>
> * Navigation (VLN-CE-Isaac): ```With the windows on your right walk forward between the windows and the patio couch. Walk towards the table and into the living room through the sliding glass doors and take a right into the dining room stopping just past the wall on the left next to the door.```
>
> Quantifying the long-horizon nature of LIBERO-Long and VLN-CE-Isaac, our table shows that they require 3.2x and 14.7x more steps to complete, respectively, compared to the short-horizon LIBERO-Spatial tasks.
>
> **(2) Description of our domain-shift experimentation:** We also conduct new experimentation to test GLUESTICK’s applicability to a VLM on a non-robotics benchmark. Specifically, we evaluate Qwen2-VL-7B-Instruct [1] using the DocVQA [2] benchmark, which assesses a VLM’s ability to answer questions from document images spanning a variety of industries (e.g., Drug, Food, Chemical).
>
> **(3) Results showcasing GLUESTICK’s generalizability to long-horizon tasks and domain shift:** As shown in the table below, across the long-horizon LIBERO-Long and VLN-CE-Isaac tasks, as well as the domain-shift setting of DocVQA, we observe that GLUESTICK generalizes well and is able to recover performance in all scenarios.
>
> | **Benchmark**     | **Average Steps** | **GLUESTICK-500's Impact**|
> |-------------------|-------------------|-|
> | LIBERO Spatial | 122.2            |Recovers 61.5% of the lost success rate
> | LIBERO Long    | 391.1            | Recovers 24.3% of the lost success rate
> | VLN-CE-Isaac    | 1828.2            | Recovers 100% of the lost success rate
> | DocVQA |1| Recovers 96% of the lost performance |
>
> *(Results in this table are calculated from Table 2 and Table 6 of our main paper.)*
>
> [1] Wang, P., Bai, S., Tan, S., Wang, S., Fan, Z., Bai, J., ... & Lin, J. (2024). Qwen2-vl: Enhancing vision-language model's perception of the world at any resolution. arXiv preprint arXiv:2409.12191.
>
> [2] Mathew, M., Karatzas, D., & Jawahar, C. V. (2021). Docvqa: A dataset for vqa on document images. In Proceedings of the IEEE/CVF winter conference on applications of computer vision (pp. 2200-2209).
>
> ---
>
> > **[W2]** The rank scheduling is empirical: with a single global r used all-way, given large per-layer variation, and the vision backbone is sensitive parameter-wise, without considering the inner difference of different layers.
>
> The standard pruning algorithms (e.g., Wanda, Magnitude, SparseGPT [1]) and quantization methods (e.g., AWQ [2], GPTQ [3]) require several hyperparameters. On the other hand, GLUESTICK is intentionally designed with only a *single* interpretable hyperparameter, $r$. Our design choice enables practitioners to easily tune the method to their specific use case by balancing model performance and safety (as detailed in Section 3.2 in our main paper).
>
> We find that the reviewer's suggestion is an excellent idea. Some practitioners may be willing to increase the algorithm's complexity through per-layer variations, for the sake of further memory reductions. We are inspired by the idea of developing a new algorithm that automatically selects the per-layer ranks. While these improvements are outside the scope of the present work, they represent a promising direction for future research. We have updated our paper's Future Work section (Section 7) accordingly based on the reviewer's suggestion.
>
> [1] Frantar, E., & Alistarh, D. (2023, July). Sparsegpt: Massive language models can be accurately pruned in one-shot. In International conference on machine learning (pp. 10323-10337). PMLR.
>
> [2] Lin, J., Tang, J., Tang, H., Yang, S., Chen, W. M., Wang, W. C., ... & Han, S. (2024). Awq: Activation-aware weight quantization for on-device llm compression and acceleration. Proceedings of machine learning and systems, 6, 87-100.
>
> [3] Frantar, E., Ashkboos, S., Hoefler, T., & Alistarh, D. (2022). Gptq: Accurate post-training quantization for generative pre-trained transformers. arXiv preprint arXiv:2210.17323.

---

> ### Author Response · Authors · 2025-11-28
> **Response by Authors [Part 3]**
>
> > **[W3]** The requirement of dense-weights: the method needs the original dense checkpoint to compute the gap SVD, which might constrain its usage in some scenarios.
>
> GLUESTICK requires access to the original dense model checkpoint, but this assumption is safely made because GLUESTICK operates alongside a pruning algorithm. It is standard across the pruning literature to require access to the original dense model. Therefore, we are taking advantage of this pre-existing assumption within GLUESTICK. Importantly, GLUESTICK does not rely on any assumptions other than those commonly adopted in the pruning literature.
>
> ---
>
> > **[Q4]** … pruning methods are highly sensitive to the quality and relevance of calibration data, and that "more data" is not always better?
>
> Although a smaller calibration set yielding slightly higher performance might seem counterintuitive, it aligns well with prior machine learning literature that highlights the importance of high-quality data. Machine learning models generally benefit more from carefully curated, high-quality datasets than from large quantities of noisy data [1]. Similarly, we find that pruning methods also perform best when provided with high-quality calibration data specifically curated for manipulation tasks, rather than simply adding more data of similar type to the existing dataset.
>
> In Appendix D.1, we showed that using a 3.5K curated manipulation calibration dataset provided a 2% performance gain over a 15K randomly selected calibration set. To provide the reviewer with additional insights, we conducted further experiments to evaluate this observation. Specifically, we explored whether an even smaller, more curated calibration dataset could improve performance. To do this, we applied K-means to cluster the states of our manipulation dataset. Based on a silhouette analysis, we selected k = 7 clusters. We then sampled 1K states from a cluster and evaluated its performance on the LIBERO Spatial benchmark. Our results are shown below:
>
>
> | **Calibration Dataset**     | **Δ Change to 15K Baseline** |
> |-----------------------------|-------------------------------|
> | 15K States (Baseline)   | 0%                            |
> | Cluster #5 (1K States)  | –1.6%                         |
> | Cluster #1 (1K States)  | +2.8%                         |
> | Cluster #0 (1K States)  | +6.8%                         |
>
> This goes to strengthen our observation that the quality of the calibration dataset could be more influential than the quantity of the dataset. It is worth noting that this results in only marginal performance improvements in our experience. We agree with the reviewer that an exciting area for future work is how the selection of calibration data can further improve the performance and safety of methods including GLUESTICK.
>
> [1] Mazumder, M., Banbury, C., Yao, X., Karlaš, B., Gaviria Rojas, W., Diamos, S., ... & Janapa Reddi, V. (2023). Dataperf: Benchmarks for data-centric ai development. Advances in Neural Information Processing Systems, 36, 5320-5347.

---

### Official Review · Reviewer_jWTz · 2025-11-03

**Soundness:** 3
**Presentation:** 4
**Contribution:** 2
**Rating:** 2
**Confidence:** 4

**Summary:**

The manuscript presents a study of VLA pruning, demonstrating that VLA models considerably lose performance compared to their LLM counterparts. The authors demonstrate and benchmark this behavior on manipulation as navigation tasks using the OpenVLA, WorldVLA, and NaVILA models. By analyzing the spectrum of model weights, the authors further demonstrate a difference in the weight space between VLAs and LLMs.

Based in this observation, GLUESTICK is proposed as a mitigation method. By compressing the most important components of the pruned weights using SVD, and thus, reconstructing the suppressed component of weights, the authors are able to recover part of the model performance. This behavior is demonstrated on simulation benchmarks for manipulation and navigation tasks.

**Strengths:**

- The analysis of performance loss is well executed, spanning multiple models and tasks.
- The proposed recovery method is grounded in the joint findings from a study considering the pruning process itself, as well as from evaluations in a robotic simulator.
- GLUESTICK is able to recover part of the lost model performance in both deployment scenarios across model architectures.
- The method is straightforward to implement without requiring target domain calibration data and shows strong improvements over the baseline.

**Weaknesses:**

- As a major shortcoming, the work misses a comparison to VLM pruning, where the strong performance drop of VLMs compared to LLM models is a known property [1,2]. In previous works, this behavior is especially prominent at or below 50% sparsity, the operating point chosen in this work. Since VLAs typically build on top of VLMs, rather than on a language-only LLM, this comparison and discussion of related works is required.
- Since VLMs lose considerable performance from pruning, it remains unclear if the demonstrated problem is a problem stemming from the VLM backbone or the full VLA built for the robotic task.
- The experiment in Q6 should be put in context. 200 SVD components are an extremely strong compression of the weight space, especially when compared to 200/500 residual components in GLUESTICK.
- Experiments are purely performed in simulation. Since real-world deployment of VLA policies can show considerably different performance, a small study on real robots can help the experiments in this work.

[1] Liang, Yinan, et al. "Efficientllava: Generalizable auto-pruning for large vision-language models." Proceedings of the Computer Vision and Pattern Recognition Conference. 2025.

[2] Koike-Akino, Toshiaki, Jing Liu, and Ye Wang. "$\mu $-MoE: Test-Time Pruning as Micro-Grained Mixture-of-Experts." arXiv preprint arXiv:2505.18451 (2025).

**Questions:**

- How strong is the performance loss at different operating points of pruning, especially with lower sparsity?
- How does Figure 2 change when compared to the corresponding VLM model?
- Overall, the work should discuss the relation to VLM pruning and put the findings and novelty in the context of existing work in this area. I will reconsider my rating if this shortcoming is adequately addressed in the paper discussion, existing works, and experimental validation.

---

> ### Author Response · Authors · 2025-11-28
> **Response by Authors [Part 1]**
>
> We appreciate the reviewer's recognition of the strengths of our work, including the breadth of our experiments across models and tasks and the practical value of GLUESTICK, which requires no calibration data yet provides substantial improvements over baselines.
>
> Below, we address each comment and summarize the additional experiments we conducted in response to the reviewer’s helpful feedback. All updates have been incorporated into the revised main paper and appendix. We also thank the reviewer for their willingness to reconsider their rating; we hope the new experiments, expanded analyses, and revised discussion fully address the reviewer’s requests.

---

> ### Author Response · Authors · 2025-11-28
> **Response by Authors [Part 2]**
>
> > **[W1]** ... the work misses a comparison to VLM pruning, where the strong performance drop of VLMs compared to LLM models is a known property … Since VLAs typically build on top of VLMs, rather than on a language-only LLM, this comparison and discussion of related works is required.
>
> > **[Q3]** Overall, the work should discuss the relation to VLM pruning and put the findings and novelty in the context of existing work in this area. I will reconsider my rating if this shortcoming is adequately addressed in the paper discussion, existing works, and experimental validation.
>
> Thank you for these constructive questions. We summarize our (1) clarifications, (2) new experiments, and (3) expanded related work below.
>
> **(1) Clarifying our main contribution and relationship to VLM pruning**
>
> Please consider that our contribution focuses on a new, robotics-driven question: *Do pruning behaviors observed in VLMs extend to robotic VLAs, and how does pruning affect safety in embodied decision-making?*
>
> While VLAs inherit components from VLMs, their perception–action coupling introduces failure modes that differ substantially from pure vision-language reasoning. Our findings complement the VLM pruning literature by showing:
>
> * VLAs are far more severely harmed by pruning than VLMs at the same sparsity levels (see new experiments below).
>
> * Pruning has major safety implications in embodied robotics—an aspect not addressed in prior pruning studies but essential for real-world deployment (e.g., see Table 1 in our main paper).
>
> **(2) New Experiments and Direct Comparisons to VLM Pruning**
>
> In response to the reviewer, we conducted two new experiments comparing VLM and VLA pruning behavior.
>
> **New Experiment A. VLM Pruning Results with GLUESTICK (New Table 6)**
>
> We applied Wanda, one of the latest state-of-the-art pruning methods, to Qwen2-VL-7B-Instruct [1] and evaluated 1,000 DocVQA examples [2]:
>
> |Base Model| **Method**                 | **Full Dense** | **30%** | **50%** | **70%** |
> |--|---------------------------|----------------|---------|---------|---------|
> |Qwen2-VL-7B-Instruct (VLM) | Wanda                 | 93.88          | 93.38   | 88.16   | 21.30   |
> |Qwen2-VL-7B-Instruct (VLM) | Wanda + GLUESTICK-500 | No Need    | No Need | 92.95   | 90.89   |
>
> **Key observations:**
>
> * At 30% sparsity, Wanda minimally affects VLM accuracy.
> * At 50% sparsity (the setting central to our VLA study due to 2:4 structured CUDA kernels), VLMs lose only ~5 ANLS points. By contrast, VLAs experience a more severe loss of ~53% success rate (see Table 5 in our main paper).
> * At 70% sparsity, GLUESTICK restores VLM performance from 21.30 to 90.89, demonstrating the strong generality of our method beyond VLAs.
>
> These results reinforce the reviewer’s point that pruning degrades VLMs, while highlighting our paper’s central finding: VLAs degrade far more dramatically, and that this has critical safety consequences. These new results are added in Table 6 and Section 5.2 of our revised main paper.
>
> **New Experiment B. Expanded Spectral Analyses Including VLMs (New Figure 2)**
>
> We extended our spectral analysis (previously LLM vs. VLA) to include VLMs. The updated Figure 2 shows:
>
> * Pruning disproportionately distorts action-critical subspaces in VLAs.
> * VLM spectra closely resemble LLM spectra and differ from VLA spectra.
>
> This mechanistic distinction explains why VLAs collapse more sharply than VLMs despite sharing similar components. Details are included in the revised main paper (Figure 2, Section 3.1).
>
> **(3) Updated Related Work (Section 2)**
>
> We expanded the Related Work section to more clearly situate our contributions relative to VLM pruning studies. The revision now:
>
> * Summarizes what prior work establishes about pruning in VLMs
> * Explains the need to evaluate whether these pruning effects are worsened in VLAs
> * Motivates the necessity of safety-oriented evaluation in robotics.
>
> We thank the reviewer again for their constructive feedback. With the new experiments, expanded analysis, and revised discussion, we hope we have fully addressed the requested comparisons to VLM pruning and clarified the novelty and significance of our work.
>
> [1] Wang, P., Bai, S., Tan, S., Wang, S., Fan, Z., Bai, J., ... & Lin, J. (2024). Qwen2-vl: Enhancing vision-language model's perception of the world at any resolution. arXiv preprint arXiv:2409.12191.
>
> [2] Mathew, M., Karatzas, D., & Jawahar, C. V. (2021). Docvqa: A dataset for vqa on document images. In Proceedings of the IEEE/CVF winter conference on applications of computer vision (pp. 2200-2209).

---

> ### Author Response · Authors · 2025-11-28
> **Response by Authors [Part 3]**
>
> > **[W2]** Since VLMs lose considerable performance from pruning, it remains unclear if the demonstrated problem is a problem stemming from the VLM backbone or the full VLA built for the robotic task.
>
> > **[Q2]** How does Figure 2 change when compared to the corresponding VLM model?
>
> We have updated Figure 2 by adding spectral results for a widely used VLM (Qwen2-VL-7B-Instruct). We find that the VLM spectra closely resemble the spectra of the LLM, both distinct from the spectra of the VLA. This supports our hypothesis that VLAs are disproportionately harmed by pruning when compared to LLMs and VLMs.
>
> ---
>
> > **[W3]** The experiment in Q6 should be put in context. 200 SVD components are an extremely strong compression of the weight space, especially when compared to 200/500 residual components in GLUESTICK.
>
> To thoroughly address this question, we ran additional experiments replacing VLA weights with low-rank SVD approximations across a range of ranks. All results were computed without pruning:
>
> | **Rank** | **200** | **400** | **500** | **800** | **1000** | **2000** | **2500** |
> |---------|---------|---------|---------|---------|----------|----------|----------|
> | Success Rate (%) | 0 | 0 | 0 | 0 | 0 | 0.4 | 21.0 |
>
> These results show that directly compressing the dense weights using SVD removes essential task-relevant structure, causing the model to fail. Our interpretation is that pruned weight matrices retain critical structure that SVD-based corrections can exploit. This offers one explanation as to why our method starts from a pruned model rather than a purely low-rank one. This extended ablation study is added in Section 5.2 and Appendix D.6 of our revised main paper.
>
> ---
>
> > **[Q1]** How strong is the performance loss at different operating points of pruning, especially with lower sparsity?
>
> We conducted additional experiments across other sparsity levels. The table below reports the impact of pruning the language backbone of OpenVLA using Wanda on the LIBERO Spatial benchmark. These results show that GLUESTICK improves performance across sparsities. However, our paper’s focus remains on 50% 2:4 structured sparsity, which is most useful in robotics due to CUDA kernel support. These new results are added in Table 5 and Section 5.2 of our revised main paper.
>
> |Base Model| **Method**                 | **0%** | **30%** | **50%** | **70%** |
> |-|---------------------------|--------|---------|---------|---------|
> |OpenVLA (VLA)| Wanda                 | 85.2   | 80.8    | 31.8    | 1.2     |
> |OpenVLA (VLA)| Wanda + GLUESTICK-500 | No Need      | 85.8    | 60.2    | 24.6    |   |
>
> ---
>
> > **[W4]** Experiments are purely performed in simulation … a small study on real robots can help …
>
> Our initial draft focused on simulation due to safety concerns, as pruning can introduce failure modes that risk damaging the robot or its environment. In response to the reviewer’s suggestion, we conducted a new real-world experiment using a Boston Dynamics Spot quadruped in a real office environment. We measure the trajectory deviation of the pruned VLA model relative to the dense VLA model. Our study demonstrates that the GLUESTICK-500 corrected pruned VLA model improved in  performance by 50%. We include our results, corresponding deployment images, and experimental overview in Figure 5 and Section 5.2 of our revised main paper.
>
> | **Metric**                | **Full Dense** | **Full Pruned** | **GLUESTICK-500** |
> |---------------------------|----------------|------------------|--------------------|
> | Trajectory Deviation (↓) | 0.00           | +10.26           | **+5.12**          |

---

### Author Response · Authors · 2025-11-28
**Overall Response to Reviewers and Area Chair**

We thank all of the reviewers (jWTz, yHH6, sjPP, aTbD) for their valuable time, careful reading, and thoughtful feedback on our work. Across the reviews, there is clear agreement that the paper tackles a real and timely problem: pruning for VLAs in robotics. Standard pruning techniques, when directly applied to VLA policies, can result in a collapse in both success and safety. The reviewers note that this behavior had not been systematically studied before, and that it is a meaningful empirical observation about how VLAs differ from LLMs.

---

**Positive Points Highlighted by Reviewers:**

* **Strong empirical demonstration across architectures and tasks.** Reviewers highlighted that the experiments are well executed and clearly show both the collapse caused by pruning and the recovery achieved by GLUESTICK across multiple architectures and tasks (jWTz, yHH6, sjPP, aTbD).

* **GLUESTICK is simple, training-free, and pruning-agnostic.** Reviewers highlighted that GLUESTICK is easy to implement and requires no additional training or calibration data (yHH6, sjPP, aTbD). It integrates cleanly into existing pruning pipelines, requires only a single SVD between dense and pruned weights, and exposes an interpretable single-rank hyperparameter to navigate the efficiency–performance tradeoff. Reviewers found these properties appealing for practical deployment in robotics (yHH6, sjPP).

* **Innovative low-rank SVD stitching mechanism.** The use of a low-rank SVD stitch back in weight space is called innovative and well motivated, and is seen as tightly connected to the empirical behavior we study (jWTz, yHH6, aTbD).

* **Insightful spectral analysis comparing VLAs and LLMs.** Reviewers appreciated the spectrum analysis that compares VLAs and LLMs, describing it as a thoughtful diagnosis that offers a plausible explanation for why VLAs are more fragile under structured pruning and as a useful guide for future work on compression and deployment (yHH6, jWTz, sjPP, aTbD).

* **Clear writing and strong presentation.** Reviewers found the paper clearly written and well presented, with a strong introduction and related work and helpful code-style illustrations of the method (jWTz, yHH6, sjPP, aTbD).

---

**New Main Experiments and Analyses Added During Rebuttal:**

To further strengthen the paper and address reviewer questions, we have added the following new main results and analyses in the revised paper (highlighted in blue):

* **VLM pruning Results with GLUESTICK (New Table 6):** VLMs degrade much less than VLAs under pruning, and GLUESTICK restores VLM performance across sparsity levels, demonstrating the strong generality of GLUESTICK beyond VLAs.

|Base Model| **Method**                 | **Full Dense** | **30%** | **50%** | **70%** |
|--|---------------------------|----------------|---------|---------|---------|
|Qwen2-VL-7B-Instruct (VLM) | Wanda                 | 93.88          | 93.38   | 88.16   | 21.30   |
|Qwen2-VL-7B-Instruct (VLM) | Wanda + GLUESTICK-500 | No Need    | No Need | 92.95   | 90.89   |

* **Expanded spectral analyses including VLMs (New Figure 2):** VLM spectra closely resemble LLM spectra and differ from VLA spectra, supporting our explanation for VLA fragility under structured pruning.

* **Real-robot experiments on Boston Dynamics Spot (New Figure 5):** GLUESTICK restores navigation abilities in a real office environment, whereas the pruned VLA exhibits poor performance.

* **Varying pruning sparsity levels for GLUESTICK (New Table 5):** GLUESTICK improves success rates across a wide range of sparsity levels, including very high sparsity.

|Base Model| **Method**                 | **0%** | **30%** | **50%** | **70%** |
|-|---------------------------|--------|---------|---------|---------|
|OpenVLA (VLA)| Wanda                 | 85.2   | 80.8    | 31.8    | 1.2     |
|OpenVLA (VLA)| Wanda + GLUESTICK-500 | No Need      | 85.8    | 60.2    | 24.6    |

* **Combining GLUESTICK with quantization (New Appendix D.4):** GLUESTICK remains compatible with other compression techniques and recovers performance even after both quantization + pruning.

| **Method** | **Dense** | **Dense & Int8** | **Sparse** | **Sparse & Int8** | **Sparse & Int8 & GLUESTICK** |
|------------|-------|------------|-------|------------|----------------------|
| Success Rate (%) | 85.2 | 84.8 | 31.2 | 31.8 | 62.2 |

* **FLOP analysis (New Appendix D.7):** The rank parameter $r$ provides a practical latency–performance tradeoff, and GLUESTICK adds minimal compute overhead relative to the savings from pruning.

| **Method**                        | **Total FLOPs (T)** |
|----------------------------------|----------------------|
| Full Dense                   | 29.32               |
| 50% 2:4 Sparse Lang. BB             | 16.54               |
| 50% 2:4 Sparse Lang. BB + GLUESTICK-200 | 18.51           |
| 50% 2:4 Sparse Lang. BB + GLUESTICK-500 | 21.47           |

*(Lower FLOPs is better)*

---

### Author Response · Authors · 2025-12-03
**Summary for Area Chair**

Dear Area Chair,

Thank you for your valuable time in reviewing our work. Below, we provide a concise summary of our contributions, our method’s strengths, our reviewers’ comments, and how our revisions and additional experiments directly address all of the reviewers' questions and comments.

---

# **TL;DR for the Area Chair**

**Contribution Summary:**

* We surprisingly find that when standard pruning methods are applied to VLAs, their task success degrades to near 0% and their safety violations significantly increase across manipulation and navigation domains.

* We study why pruning fails specifically in VLAs, as compared to LLMs and VLMs, based on our spectral analysis.

* We introduce GLUESTICK, a simple, training-free, pruning-agnostic SVD-based correction that substantially recovers task success and safety while preserving the sparsity benefits of pruning.

**Strength Summary:**

* **General:** GLUESTICK consistently restores performance across VLA models, robotic tasks, sparsity levels, real-robot evaluations, VLMs, domain shifts, and long-horizon scenarios.

* **Practical:** GLUESTICK is training-free, pruning-agnostic and uses no additional training or calibration data.

* **Efficient:** GLUESTICK preserves the memory and latency benefits of pruning and is fully compatible with other compression methods such as quantization.

**Reviewer and Rebuttal Summary:**

* **Reviewer aTbD (Score: 10)** strongly endorses the paper as sound and impactful and recommends it as a highlight. The reviewer notes that it addresses the key problem of making VLA models fast, highlights the importance of our pruning-induced collapse discovery, and regards GLUESTICK as an effective, pruning-agnostic low-rank recovery method.

* **Reviewer sjPP (Score: 6)** asked about the compatibility and latency implications of GLUESTICK. Through new experiments, we show that GLUESTICK is fully compatible with quantization and, based on a FLOP analysis, preserves the latency benefits of pruning.

* **Reviewer yHH6 (Score: 4)** raised questions about our focus on linear layers, as well as about GLUESTICK’s performance under domain shift and long-horizon tasks. Through new analysis and experiments, we show that linear layers constitute the vast majority of parameters in VLA models (93-98%), and we demonstrate that GLUESTICK generalizes to domain shift and long-horizon scenarios.

* **Reviewer jWTz (Score: 2)** primarily asked for a clearer comparison between VLA and VLM pruning as well as validating GLUESTICK on a real robot. **They explicitly wrote that they will reconsider their rating if this was adequately addressed.** Our revision responds directly with (1) new experiments showing that pruning harms VLAs far more severely than VLMs, (2) new evaluations demonstrating GLUESTICK recovers VLM performance on additional benchmarks, and 3) new real-world robotic hardware results to further validate effectiveness.

Rebuttals were submitted with sufficient time for discussion; however, the discussion period closed before reviewers could respond. **Please find a summary of all of our new experiments and their strong results in the previous overall message.** We are optimistic that by directly addressing every reviewer comment with rigorous new experimentation, we would exceed the reviewers’ expectations. Thank you for your consideration.

Best regards,

Authors of Paper #21771

---

### Meta-Review · Area_Chair_2i1d · 2026-01-07

**Summary:**

The paper "Don't Run with Scissors: Pruning Breaks VLA Models but They Can Be Recovered" reports that standard structured pruning (Magnitude, Wanda) can collapse Vision-Language-Action policies: OpenVLA and NaVILA drop to 0% success, while unsafe-episode rates increase (up to 100% on NaVILA). It attributes this to VLA layers having flatter singular-value spectra than LLM/VLM layers, so pruning removes useful signal spread across many directions. It proposes GLUESTICK: compute a truncated SVD of the dense-minus-pruned weight gap and add a low-rank correction at inference, with no retraining. Strengths are the clear failure case and a simple recovery, including a Spot robot demo. However, novelty is limited (low-rank add-on), it requires the dense checkpoint, and rank choice is still heuristic.

**Reviewer Concerns:**

Addressed: added VLM comparisons and expanded spectral analysis, added real-robot Spot experiment, and broadened sparsity/efficiency evidence (including a rank dial and discussion of per-layer rank scheduling as future work).

Still outstanding: limited conceptual novelty vs other low-rank corrections, dependence on having dense weights, no principled per-layer rank rule, and weak wall-clock latency reporting (mostly analytic/indirect).

**Reviewer Scores:**

aTbD: stays at 10.
sjPP (6): likely improve after added quantization/efficiency evidence, but still cautious on novelty.
yHH6 (4): likely improve  given added ablations and generalization claims, but still flags rank scheduling and dense-checkpoint dependence.
jWTz (2): likely 2 -> 4 if VLM comparisons and Spot results are in the revision, but may still reject on novelty/positioning.

---

### Decision · Program_Chairs · 2026-01-26

Reject